# REPRESENTATION-ALIGNED MULTI-SCALE PERSONALIZATION FOR FEDERATED LEARNING

## ABSTRACT

In federated learning (FL), accommodating clients with diverse resource constraints remains a significant challenge. A widely adopted approach is to use a shared full-size model, from which each client extracts a submodel aligned with its computational budget. However, regardless of the specific scoring strategy, these methods rely on the same global backbone, limiting both structural diversity and representational adaptation across clients. This paper presents FRAMP, a unified framework for personalized and resource-adaptive federated learning. Instead of relying on a fixed global model, FRAMP generates client-specific models from compact client descriptors, enabling fine-grained adaptation to both data characteristics and computational budgets. Each client trains a tailored lightweight submodel and aligns its learned representation with others to maintain global semantic consistency. Extensive experiments on vision and graph benchmarks demonstrate that FRAMP enhances generalization and adaptivity across a wide range of client settings.

## 1 INTRODUCTION

Federated learning (FL) has emerged as a promising paradigm for distributed model training, enabling multiple clients to collaboratively learn a shared model without exchanging raw data (Konečnỳ, 2016). In this setting, a central server coordinates the training by aggregating locally updated models from clients. However, FL faces challenges in real-world deployments due to system heterogeneity, where clients differ significantly in computational capabilities (Diao et al., 2020; Lin et al., 2020). A practical solution is submodel extraction, which enables model heterogeneity by allowing each client to train a sparse subnetwork derived from a shared full-size model.

Static submodel extraction (Diao et al., 2020; Kim et al., 2022; Ilhan et al., 2023; Kang et al., 2023; Wang et al., 2023; Yi et al., 2024) determines each client's submodel architecture before training and keeps it fixed throughout the entire process. However, such static designs fail to account for the evolving model dynamics, often resulting in suboptimal performance. In contrast, dynamic submodel extraction (Caldas et al., 2018; Horvath et al., 2021; Alam et al., 2022; Liao et al., 2023; Chen et al., 2023; Wang et al., 2024; Wu et al., 2024) updates submodel structures across rounds to better track model evolution. For example, FedRolex (Alam et al., 2022) employs a rolling strategy that treats all parameters equally, while FIARSE (Wu et al., 2024) selects submodels based on parameter importance scores.

Existing methods support training under diverse resource constraints but often rely on a single shared full-size model for submodel extraction. This shared backbone overlooks client-specific characteristics, leading to structurally similar submodels and performance disparities across clients, as shown in Figs. 1 and 2. Additionally, data heterogeneity presents a critical challenge, as clients with non-IID data may develop divergent representations, resulting in poor generalization and semantic misalignment across the federation.

In this paper, we revisit submodel personalization in FL by addressing two key questions:

**Q1** **How can we construct personalized submodels that adapt to both the computational constraints and data distribution of each client?**

**Q2** **How can we ensure semantic consistency across clients without relying on any shared public dataset?**

To address **Q1**, we propose client-aware full-size model generation, where each client is assigned a personalized full model conditioned on its compact descriptor. This approach eliminates reliance on a fixed global backbone and reduces redundancy, as many parameters in a shared global model may be irrelevant to specific clients. Building on this, we implement adaptive submodel extraction, where submodels are dynamically selected based on evolving parameter values during training, implicitly capturing parameter importance without additional tracking.

To address **Q2**, we introduce prototype-guided representation alignment, which promotes semantic consistency by aligning class-level prototypes across clients during local training, without requiring any shared public dataset.

These components form a unified framework that supports personalization in both system-level resource constraints and data heterogeneity.

We call our framework *Federated Representation-Aligned Multi-scale Personalization (FRAMP)*. FRAMP preserves the simplicity of centralized model management by encoding personalization into a single global generator. Our main contributions are summarized as follows:

- We propose a client-aware model generation mechanism that leverages compact client descriptors to instantiate personalized full-size models, enabling submodel extraction tailored to both computational and data characteristics.

- We introduce an adaptive submodel extraction strategy that dynamically encodes parameter importance within the training process, allowing submodels to evolve without additional overhead.

- We incorporate a prototype-guided representation alignment strategy that ensures semantic consistency across clients, without requiring any shared public datasets.

- Extensive experiments demonstrate that FRAMP achieves robust personalization and generalization, maintaining strong performance even under extreme heterogeneity and unseen client scenarios.

## 2 RELATED WORK

This section highlights related research directions, with a more extensive review in Appendix A.

### 2.1 MODEL HETEROGENEITY IN FL

FL systems often encounter computational heterogeneity, where clients differ significantly in hardware capabilities and resource budgets. A common strategy is to assign more local updates to high-resource clients while limiting those for resource-constrained ones (Li et al., 2020b; Mitra et al., 2021; Shin et al., 2022; Wu et al., 2023). However, such approaches generally assume that all clients are capable of training the full-size model, which is impractical for severely constrained devices. To address this, several works explore model customization (Lin et al., 2020; Zhang et al., 2021; Cho et al., 2022; Zhang et al., 2023; Wu et al., 2024), enabling clients to train local models adapted to computational capacity. Aggregation across heterogeneous models is often performed via knowledge distillation, which typically requires a shared public dataset (Alballa & Canini, 2023). However, such requirements limit applicability in decentralized environments where public data is unavailable.

Another approach is model sparsification, where clients prune less important parameters to meet local resource constraints (Chen et al., 2023; Liao et al., 2023; Zhou et al., 2023). While effective, these methods may introduce significant computational and memory overhead, limiting their practicality in low-resource scenarios. In contrast, our framework supports resource-adaptive submodel training without requiring public data and encodes parameter utility implicitly through their values. Furthermore, by coupling model sparsity with personalization, our approach provides a unified solution to both system and data-level heterogeneity, which is rarely addressed simultaneously in existing works.

### 2.2 MODEL PERSONALIZATION IN FL

To address data heterogeneity in FL, a wide range of personalized federated learning (PFL) approaches have been developed. These include local fine-tuning (Schneider & Vlachos, 2021), regularization-based objectives (Hanzely et al., 2020; Yan et al., 2024), model mixing (Ma et al., 2022), meta-learning (Jiang et al., 2019; Lee et al., 2024), parameter decompositions (Arivazhagan et al., 2019). Other works explore server-side personalization by clustering clients and maintaining multiple global

models for different groups (Ghosh et al., 2020; Huang et al., 2021), or by fully decoupling the training of individual models with periodic collaboration (Zhang et al., 2021; Ye et al., 2023; Scott et al., 2024; Liang et al., 2025). However, most PFL approaches assume structurally identical models across clients, overlooking constraints imposed by device heterogeneity. In contrast, our framework jointly enables resource-aware model adaptation and personalization through a unified architecture.

# 3 PRELIMINARIES

## 3.1 PROBLEM FORMULATION

We consider an FL scenario involving $N$ clients and a central server. Each client $n \in [N]$ has access to a local dataset $\mathcal{D}_n = \{(x_i^n, y_i^n)\}_{i=1}^{K_n}$, which may follow a unique, client-specific distribution. The computational capacity of client $n$ is defined by a sparsity budget $\gamma_n \in [0, 1]$, indicating the maximum fraction of a full-size model $f_n \in \mathbb{R}^d$ that the client can utilize. This budget $\gamma_n$ varies across clients to reflect their differing resource constraints.

Each client $n$ selects a binary mask $\mathcal{M}_n \in \{0, 1\}^d$ to extract its submodel, subject to the constraint $\|\mathcal{M}_n\|_1 \leq \gamma_n d$. Let $\mathcal{M} = \bigcup_{n \in [N]} \mathcal{M}_n \in \{0, 1\}^{N \times d}$ denote the collection of all clients' masks. To jointly optimize the model parameters and the masks, the overall objective is defined as:

$$\underset{\{f_n\}, \mathcal{M}}{\arg\min} \frac{1}{N} \sum_{n=1}^{N} \mathbb{E}_{(x,y) \sim \mathcal{D}_n}[\mathcal{L}_n((f_n \odot \mathcal{M}_n)(x), y)], \tag{1}$$

where $\mathcal{L}_n$ represents the task-specific loss computed on client $n$'s local dataset $\mathcal{D}_n$, and $\odot$ denotes the elementwise product. For simplicity, we assume uniform client weighting, though the formulation can naturally extend to non-uniform weights.

## 3.2 LIMITATIONS OF EXISTING APPROACHES

Many existing methods rely on a shared full-size global model, where all clients use the same parameters, i.e., $f_1 = f_2 = \cdots = f_N$, from which they extract their local submodels. Each client applies a binary mask $\mathcal{M}_n$ to select a subset of parameters for local training. These approaches can be categorized into two types: static masking, where the mask remains fixed throughout the training process, and dynamic masking, where the mask is updated across FL training rounds.

Static methods assign fixed masks to clients, while dynamic approaches allow masks to adapt during training. Some recent methods incorporate importance scores to guide mask selection (Liao et al., 2023; Chen et al., 2023), alternating between optimizing these scores and updating model parameters. However, such designs often introduce significant computational and storage overhead. Separating parameter updates and mask optimization into distinct steps can also be inefficient. FIARSE (Wu et al., 2024) mitigates this by ranking the parameters based on their magnitude and pruning accordingly, avoiding explicit mask optimization. However, it still relies on a globally shared full model and applies a fixed global ranking for submodel extraction across all clients. This approach has two major drawbacks:

- **Limited Structural Diversity:** Since all clients prune from the same global full model using a universal importance ranking, the extracted submodels are structurally similar, regardless of variations in client data distributions.

- **Client-Agnostic Importance Estimation:** Parameter importance is determined without accounting for client-specific characteristics, often yielding submodels that fail to adapt to client data.

As shown in Fig. 1, the cumulative coverage plots across four sparsity levels $\gamma_n$ reveal that FIARSE concentrates mask activations on a similar subset of model parameters across clients within each model-size group. For example, the 1/64 submodels (red line) place more than 60% of mask activations within the first 20% of the parameter index range, while many later parameters have low selection probability. This behavior stems from all clients selecting submodels based on a shared importance score ranking. Although FIARSE supports dynamic submodel sampling in each round, parameters with low importance scores are seldom chosen, resulting in limited training opportunities for these parameters. Consequently, a significant portion of the model remains underutilized, leading

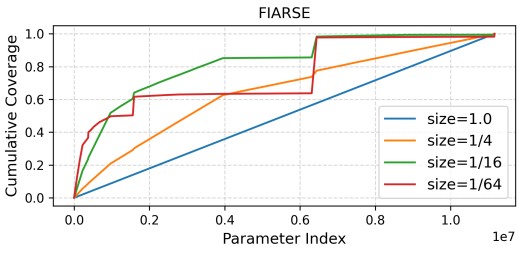 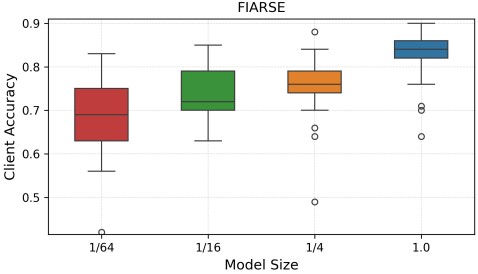

Figure 1: Cumulative distribution of preserved parameters across submodels of different sizes. Each curve represents the cumulative proportion of parameters selected (mask=1).

Figure 2: Client accuracy distribution of submodels across different sizes. Each box represents the variation in accuracy among clients.

to high structural similarity and reduced diversity among submodels. Additional discussion is in Appendix D.7.

Furthermore, Fig. 2 highlights significant performance disparities among clients, particularly those with smaller model sizes, suggesting that the extracted submodels fail to address client-specific needs. Additionally, most existing methods overlook the challenge of representation misalignment, where clients learn inconsistent class prototypes due to model heterogeneity and non-IID data (Kong et al., 2024), ultimately compromising generalization performance.

## 4 METHODOLOGY

FRAMP proceeds in three key stages, which are repeated during every communication round. An overview of the framework is shown in Fig. 3.

- **Stage 1 – Client-Aware Full-Size Model Generation** (Section 4.1): The server generates a personalized full-size model for each client $n$ by feeding its client-specific vector $\mathbf{v}_n$ into the server model $H(\cdot; \boldsymbol{\varphi})$, which outputs model parameters tailored to the client.

- **Stage 2 – Adaptive Submodel Extraction** (Section 4.2): Each client adaptively derives a sparse submodel from the generated full-size model based on importance scores, constrained by its local computation budget $\gamma_n$.

- **Stage 3 – Representation Alignment** (Section 4.3): Clients train their submodels on local data while aligning class-level representations with global prototypes to maintain semantic consistency across heterogeneous clients.

At initialization, each client $n$ uploads its vector $\mathbf{v}_n$ to the server, serving as the input for model generation. Stages 1 to 3 are then repeated until the final communication round $R$. In summary, FRAMP leverages client-specific information to generate personalized full-size models, dynamically extracts sparse submodels based on parameter importance, and promotes semantic alignment during local training. The full procedure is detailed in Algorithm 1 in Appendix B.

### 4.1 CLIENT-AWARE FULL-SIZE MODEL GENERATION

To address the limitations of global model sharing, we generate a personalized full-size model $f_n(\cdot; \boldsymbol{\omega}_n)$ for each client $n$, where the parameters $\boldsymbol{\omega}_n$ are produced by an hypernetwork (HN) $H(\cdot; \boldsymbol{\varphi})$ conditioned on a client-specific descriptor $\mathbf{v}_n \in \mathbb{R}^l$. This design enables per-client model generation in a scalable and unified framework, while remaining consistent with the general problem formulation.

At initialization, each client computes its descriptor $\mathbf{v}_n$ by aggregating embeddings of local data instances. Specifically, we employ a feature extractor $\phi$ to encode individual samples and take the average over the local dataset, i.e., $\mathbf{v}_n = \frac{1}{|K_n|} \sum_{(x,y) \in \mathcal{D}_n} \phi(x)$. The descriptor can be fixed or updated during training, provided that a meaningful client representation is available a priori.

The HN maps each client descriptor $\mathbf{v}_n$ to the corresponding model parameters:

$$\boldsymbol{\omega}_n := H(\mathbf{v}_n; \boldsymbol{\varphi}), \tag{2}$$

Figure 3: Overview of the FRAMP framework. The server generates personalized full models using HN and extracts submodels via dynamic masking for each client. Clients train these submodels on local data and align class-level prototypes with server-aggregated global prototypes to ensure semantic consistency. Updates are used to refine the HN and global prototypes.

defining the full-size model $f_n$ for client $n$. By mapping each descriptor to a personalized full-size model, the HN ensures that all client-specific models reside on a shared manifold in the parameter space (Shamsian et al., 2021). This approach allows each full-size model to adapt to the unique characteristics of its client's data while maintaining a structured parameter space that promotes generalization. Sharing the HN parameters $\varphi$ enables efficient knowledge transfer across clients, balancing personalization with collaborative learning.

## 4.2 ADAPTIVE SUBMODEL EXTRACTION

Prior studies (Mostafa & Wang, 2019; Jayakumar et al., 2020; Wu et al., 2024) have demonstrated that the magnitude of model parameters can serve as a useful proxy for their importance. By controlling the number of active parameters in a submodel, we constrain its computation and storage cost to meet client-specific budgets. To this end, we define a client-specific threshold such that only parameters with absolute values exceeding this threshold are retained in the submodel. Formally, the binary mask in Eq. (1) becomes a deterministic function of the model parameters:

$$\underset{\{f_n\}}{\arg\min} \frac{1}{N} \sum_{n=1}^{N} \mathbb{E}_{(x,y)\sim\mathcal{D}_n}[\mathcal{L}_n((f_n \odot \mathcal{M}_n(\boldsymbol{\omega}_n))(x), y)], \text{where } \mathcal{M}_n(\boldsymbol{\omega}_n) = \begin{cases} 1, & \text{if } |\boldsymbol{\omega}_n| \geq \boldsymbol{\theta}_n, \\ 0, & \text{if } |\boldsymbol{\omega}_n| < \boldsymbol{\theta}_n. \end{cases}$$
(3)

Here, $\mathcal{M}_n(\cdot)$ denotes the mask function for client $n$, determined by a threshold $\boldsymbol{\theta}_n$ such that $\|\mathcal{M}_n(\boldsymbol{\omega}_n)\|_1 \leq \gamma_n d$. The threshold $\boldsymbol{\theta}_n$ is determined according to each client's local computation budget. Specifically, we apply a $\text{TopK}_{\gamma_n}(\cdot)$ operation to retain the top $\gamma_n d$ entries in $|\boldsymbol{\omega}_n|$, where $\boldsymbol{\theta}_n$ is set as the minimum value among the selected entries.

This formulation reduces submodel extraction to a thresholding operation over parameter magnitudes. As a result, optimizing the model parameters implicitly determines the mask and forms the submodel, effectively prioritizing parameters with higher importance without requiring explicit mask updates.

## 4.3 REPRESENTATION ALIGNMENT

To enhance semantic consistency across clients, we align class-level representations derived from the local model. For this purpose, we decompose each client model $f_n$ into two components: an *encoder* $e_n : \mathbb{R}^k \to \mathbb{R}^h$, which maps inputs to $h$-dimensional latent representations, and a *prediction head* $g_n : \mathbb{R}^h \to \mathbb{R}^C$, which produces logits for $C$ classes. Each client $n$ computes a local prototype $\mathbf{P}_n^c \in \mathbb{R}^h$ for class $c$ by averaging the encoder outputs over the local samples belonging to that class as $\mathbf{P}_n^c = \frac{1}{|\mathcal{X}_n^c|} \sum_{x \in \mathcal{X}_n^c} e_n(x)$, where $\mathcal{X}_n^c$ denotes the set of client $n$'s local samples with label $c$.

After local training, each participating client uploads its set of local prototypes $\{\mathbf{P}_n^c\}_{c=1}^C$ to the server. The server aggregates these to obtain a global prototype for each class via $\mathbf{P}^c = \frac{1}{N} \sum_{n=1}^N \mathbf{P}_n^c$, and broadcasts the global prototype set $\mathbf{P}_g = \{\mathbf{P}^c\}_{c=1}^C$ back to the clients.

To leverage global semantic knowledge during local updates, a prototype alignment regularization term is added to the local loss function. This term encourages the client's local representations to stay close to the global counterparts as

$$\mathcal{L}_n = \mathcal{L}_{CE} + \lambda \mathcal{L}_R, \ \mathcal{L}_R = \sum_{c=1}^C \mathrm{dist}(\mathbf{P}_n^c, \mathbf{P}^c), \tag{4}$$

where $\mathcal{L}_{CE}$ is the standard cross-entropy loss, $\mathcal{L}_R$ is the prototype alignment loss, $\lambda$ is a balancing coefficient, and $\mathrm{dist}(\cdot, \cdot)$ is a distance function (e.g., Euclidean distance).

### 4.4 Learning Procedure

With the personalized model generation and submodel extraction mechanisms, the overall learning objective is reformulated as:

$$\arg\min_{\boldsymbol{\varphi}} \frac{1}{N} \sum_{n=1}^N \mathbb{E}_{(x,y)\sim\mathcal{D}_n}[\mathcal{L}_n((H(\mathbf{v}_n; \boldsymbol{\varphi}) \odot \mathcal{M}_n(\boldsymbol{\omega}_n))(x), y)]. \tag{5}$$

After receiving the submodel $f_n \odot \mathcal{M}_n(\boldsymbol{\omega}_n)$ from the server in the current communication round, client $n$ performs $T$ steps of local training on its private dataset $\mathcal{D}_n$, updating only the active submodel parameters as $\hat{\boldsymbol{\omega}}_n^{t+1} = \hat{\boldsymbol{\omega}}_n^t - \alpha \nabla_{\hat{\boldsymbol{\omega}}_n} \mathcal{L}_n(\hat{\boldsymbol{\omega}}_n^t)$, where $\hat{\boldsymbol{\omega}}_n$ denotes the submodel parameters selected by the mask. After $T$ steps, the client transmits the update $\Delta\hat{\boldsymbol{\omega}}_n = \hat{\boldsymbol{\omega}}_n^T - \hat{\boldsymbol{\omega}}_n^0$ to the server. The server then updates the HNs parameters $\boldsymbol{\varphi}$ as $\boldsymbol{\varphi} := \boldsymbol{\varphi} - \beta \Delta\boldsymbol{\varphi}$, where $\Delta\boldsymbol{\varphi} = (\nabla_{\boldsymbol{\varphi}} \hat{\boldsymbol{\omega}}_n)^T \Delta\hat{\boldsymbol{\omega}}_n, \nabla_{\boldsymbol{\varphi}} \hat{\boldsymbol{\omega}}_n = \mathcal{M}_n \odot \nabla_{\boldsymbol{\varphi}} \boldsymbol{\omega}_n$. This formulation allows FRAMP to jointly optimize the server-side model and client-specific submodels through end-to-end gradient-based updates.

## 5 Experiments

### 5.1 Experimental Settings

**Datasets** We evaluate FRAMP on image classification (CIFAR10, CIFAR100 (Krizhevsky et al., 2009)) and node classification on graph-structured data (ogbn-arxiv (Hu et al., 2020)). Following (Wu et al., 2024), CIFAR10 and CIFAR100 are partitioned into 100 clients using a Dirichlet distribution with concentration parameter $\alpha = 0.3$, while ogbn-arxiv is partitioned into disjoint subgraphs per client as in (Baek et al., 2023). Further partitioning details are provided in Appendix C.1.

**Baselines** We compare FRAMP with state-of-the-art FL methods supporting model heterogeneity, including static submodel extraction approaches HeteroFL (Diao et al., 2020), ScaleFL (Ilhan et al., 2023), and dynamic strategies FedRolex (Alam et al., 2022), FIARSE (Wu et al., 2024). All baselines extract submodels from a shared global backbone, without client-specific personalization.

**System Heterogeneity** To simulate system-level constraints, we vary the sparsity ratio $\gamma_n$, which specifies the proportion of the full model that client $n$ can store and train. We evaluate four capacity levels, $\gamma' = \{1/64, 1/16, 1/4, 1.0\}$, with clients evenly assigned to each level. Our framework can be easily extended to other client distributions or finer-grained capacity groups.

**Implementation Details** The client participation ratio is set to 10% by default. Training is run for 800 communication rounds on image tasks and 200 rounds on graph tasks. We use ResNet-18 for image tasks and a four-layer GCN for graph tasks. The HNs in FRAMP is implemented as a two-layer MLP. Reported results are averaged over three random seeds. All baseline results are obtained on the best hyperparameter settings as in Appendix C.2.

### 5.2 Performance Comparison

We evaluate the performance of submodels on each client's local test dataset, as summarized in Table 1. The columns labeled "1/64" to "1.0" report the average accuracy grouped by model size $\gamma_n$, while the "Local" column shows the overall average across all clients. FRAMP consistently

Table 1: Test accuracy of submodels across four sizes on CIFAR10, CIFAR100, and ogbn-arxiv.

| Method | CIFAR-10 | | | | | | CIFAR-100 | | | | | | ogbn-arxiv | |
|---|---|---|---|---|---|---|---|---|---|---|---|---|---|---|
| | Local | 1/64 | 1/16 | 1/4 | 1.0 | Union | Local | 1/64 | 1/16 | 1/4 | 1.0 | Union | Local | Union |
| HeteroFL | 68.88 | 60.24 | 69.32 | 72.18 | 73.76 | 66.05 | 31.75 | 27.24 | 29.80 | 33.52 | 36.44 | 30.67 | 31.53 | 31.26 |
| FedRolex | 67.18 | 54.60 | 64.96 | 70.08 | 79.08 | 65.98 | 31.67 | 21.00 | 30.84 | 36.44 | 38.40 | 29.89 | 28.26 | 27.98 |
| ScaleFL | 72.10 | 69.04 | 71.64 | 70.08 | 77.64 | 67.37 | 39.69 | 36.16 | 40.48 | 42.56 | 39.56 | 37.56 | 41.53 | 37.77 |
| FIARSE | 77.04 | 73.12 | 77.20 | 77.24 | 82.04 | 73.75 | 41.76 | 39.12 | 43.24 | 43.72 | 40.96 | 38.63 | 46.18 | 41.53 |
| **FRAMP** | **79.11** | **78.20** | **78.56** | **77.40** | **82.28** | **75.65** | **42.95** | **43.00** | **44.08** | **43.72** | **41.00** | **40.26** | **48.00** | **44.34** |

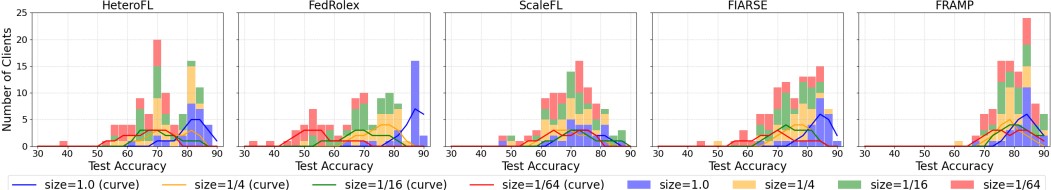

Figure 4: Histograms on CIFAR10 showing client counts at different test accuracy levels under four submodel sizes for each baseline. The curves depict the accuracy distribution for each model size.

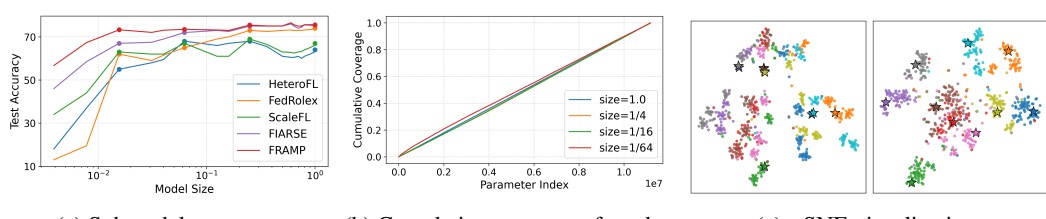

(a) Submodel accuracy.    (b) Cumulative coverage of masks.    (c) t-SNE visualization.

Figure 5: (a) Test accuracy of submodels with different sizes for all baselines, evaluated on the union test set of CIFAR-10. (b) Cumulative coverage of parameter masks for different submodel sizes in FRAMP, showing a more uniform and efficient utilization of full-model parameters across submodels, compared to Fig. 1. (c) t-SNE visualization of local prototypes (dots, one per class per client) and global prototypes (stars) in FRAMP on CIFAR10. Each color represents a class.

outperforms all baselines, with particularly notable gains for clients with limited resources. In the smallest submodel setting (1/64), FRAMP achieves substantial improvements across all datasets, suggesting that combining dynamic masking with HN-based full-model personalization effectively guides resource-constrained models to the most informative parameters.

Fig. 4 illustrates the distribution of client accuracies across different submodel sizes. FedRolex, with its rolling-based strategy, exhibits a dispersed and inconsistent distribution, highlighting challenges in optimizing local submodel performance. HeteroFL shows notable disparities across model sizes, with smaller submodels performing substantially worse than full-size models. FIARSE achieves strong results for larger models but suffers from greater variability and a pronounced long-tail effect for smaller models, indicating reduced accuracy for resource-constrained clients.

In contrast, FRAMP exhibits a compact, right-shifted distribution across all submodel sizes, reflecting consistently high accuracy regardless of model capacity. The smoothed curves for different sizes largely overlap, indicating strong inter-size consistency. Overall, FRAMP delivers balanced, stable, and robust performance across all submodel sizes, even under extreme resource constraints.

These advantages stem from FRAMP's balanced parameter utilization and representation alignment. Fig. 5b shows more even mask activations across parameters. Fig. 5c demonstrates representation alignment, with local prototypes (dots) relatively dispersed at an earlier stage (left) converging toward global prototypes (stars) at a later stage (right), forming tighter class clusters.

## 5.3 GENERALIZATION ON UNION TEST SET

Evaluating on the union test set provides a measure of global generalization beyond individual client distributions. We test submodels of different sizes on a union set formed by merging all clients' local test data. For baselines that train a global full-size model, submodels are extracted directly from

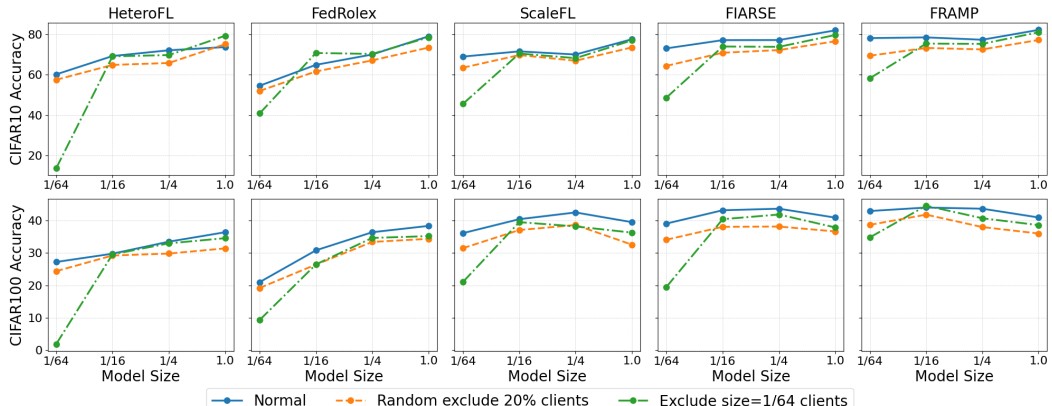

Figure 6: Test accuracy of submodels (across four model sizes) on CIFAR10 and CIFAR100, evaluated under three training settings: (1) training with all clients; (2) randomly excluding 20% of clients from each model size group; (3) excluding only clients with the smallest model size ($\gamma = 1/64$).

the trained model. Since FRAMP does not explicitly train such a model, we synthesize one via the trained HNs using the average of all client descriptors, and extract submodels in the same manner.

The "Union" column in Table 1 reports the average performance of submodels (across sizes that participated in training) evaluated on the union test set. FRAMP consistently outperforms all baselines.

We further consider a more challenging scenario where unseen clients require submodels of sizes not encountered during training. To simulate this, we extract both larger and smaller submodels and evaluate them on the union test set. As illustrated in Fig. 5a, FRAMP consistently outperforms all baselines. While larger submodels generally lead to better performance for all methods, the baselines, particularly static ones such as HeteroFL and ScaleFL, suffer significant degradation under size shifts. In contrast, FRAMP maintains strong adaptability even under extreme deviations.

## 5.4 Generalization to New Clients

While the previous section evaluates submodels on a union test set using clients that participated in training, this section focuses on generalization to entirely unseen clients. In real-world deployments, it is common for new clients with distinct data to arrive after training, necessitating personalized models despite not being involved in the optimization process.

To simulate this scenario, we randomly exclude 20% of clients from each model size group during training and evaluate the resulting models directly on their local test sets. For baselines, submodels are extracted from the trained global model and evaluated on the new clients. For FRAMP, we freeze the model generator $H(\cdot; \boldsymbol{\varphi})$ and initialize personalized models using the descriptors of the new clients. Fig. 6 shows FRAMP maintains comparable accuracy on unseen clients with minimal degradation.

We further evaluate an extreme case where all clients with model size $\gamma_n = 1/64$ are excluded from training. This setting examines generalization to capacity constraints entirely absent during optimization. As shown in Fig. 6, FIARSE exhibits a sharp performance drop on excluded $1/64$ clients, and this exclusion also adversely impacts performance on other model sizes. Although FRAMP also experiences a decline, it consistently outperforms other baselines, demonstrating strong robustness to missing capacity profiles.

## 5.5 Ablation Studies

**Impact of Model Personalization and Representation Alignment** We conduct an ablation study by selectively disabling individual modules. Table 2 reports the performance of four variants. To isolate semantic alignment, we remove prototype loss by setting $\lambda = 0$. To assess personalization, we replace HNs with a shared full-size model while retaining the same masking strategy. Finally, we remove both. The results show that disabling either module consistently reduces performance. While the average accuracy drop is modest, removing the alignment loss notably increases class-wise accuracy variance. These findings indicate that the two modules provide complementary benefits:

model personalization improves adaptability to heterogeneous clients, while prototype alignment fosters semantic consistency under non-IID distributions.

**Submodel Extraction Strategy Analysis**   We compare different strategies for extracting submodels under resource constraints. Specifically, we evaluate three approaches: (i) TopK-based masking by parameter magnitude (adopted in FRAMP), (ii) layerwise TopK masking that selects within each layer, and (iii) a learned projection method that adds an MLP to the full model output from HNs.

As shown in Table 2, global TopK masking consistently performs best across all model sizes. Layerwise TopK performs worse, since independent selection within each layer ignores global parameter importance and leads to suboptimal allocation. The MLP-based approach, though end-to-end, suffers from unstable training and degraded performance. This may be due to the difficulty of learning structured sparsity and controlling sparsity levels. In contrast, global TopK provides a simple yet effective way to enforce budget constraints while retaining essential parameters.

Table 2: Ablation study on CIFAR10.

| Method | Local | 1/64 | 1/16 | 1/4 | 1.0 | Union |
|---|---|---|---|---|---|---|
| FRAMP | 79.11 | 78.20 | 78.56 | 77.40 | 82.28 | 75.65 |
| w/o RA[a] | 74.49 | 75.16 | 74.12 | 72.44 | 76.24 | 72.40 |
| w/o Per[b] | 75.21 | 70.68 | 74.40 | 75.44 | 80.32 | 71.17 |
| w/o Both[c] | 73.97 | 68.68 | 74.40 | 74.72 | 78.08 | 70.84 |
| Layerwise[d] | 67.79 | 63.00 | 69.04 | 67.52 | 71.60 | 65.46 |
| MLP[e] | 67.12 | 62.92 | 66.96 | 65.68 | 72.92 | 64.12 |
| Onehot[f] | 76.52 | 76.40 | 77.60 | 74.04 | 78.04 | 74.27 |
| Update[g] | 78.69 | 77.02 | 78.00 | 77.40 | 82.32 | 75.01 |

[a] Remove the representation alignment loss $\mathcal{L}_R$.
[b] Replace personalized full model with a shared global model.
[c] Remove full model generation and representation alignment.
[d] Use layer-wise TopK masking instead of global TopK.
[e] Replace masking with an MLP-based submodel generator.
[f] Use onehot vectors as descriptors.
[g] Update the descriptors during training.

**Descriptor Strategy Analysis**   We compare three approaches: (i) using a randomly initialized feature extractor to generate fixed descriptors (used in FRAMP), (ii) assigning each client a one-hot vector, and (iii) a trainable feature extractor and updating descriptors every 10 communication rounds. As shown in Table 2, one-hot vectors perform worst, as lacking client-specific information. (i) and (iii) yield similar accuracy, but (iii) incurs higher client computation due to training the feature extractor.

Sensitivity of $\lambda$, controlling the weight of the alignment loss, is reported in Appendix D.4.

## 5.6   FURTHER DISCUSSIONS

**Computational Overhead**   The HNs operate entirely on the server, adding no extra computation for clients compared to standard heterogeneous FL baselines. At initialization, each client sends its descriptor to server once, which remains fixed throughout training, incurring no extra communication. The only additional cost during training is prototype exchange, which adds negligible overhead. Additional discussion is provided in Appendix D.5.

Table 3 reports the server-side training time per communication round. FRAMP introduces a modest overhead in the first round due to the initialization of the model generator, but maintains comparable training time to existing baselines in subsequent rounds.

Table 3: Server training time per communication round on CIFAR10 (in milliseconds).

| Method | HeteroFL | FedRolex | ScaleFL | FIARSE | FRAMP |
|---|---|---|---|---|---|
| Rnd 1 | 24.74 | 54.05 | 25.45 | 26.89 | 37.96 |
| Rnd 2+ | 8.01 | 8.09 | 8.35 | 7.67 | 7.98 |

**Privacy Discussion**   FRAMP transmits three types of information during training: submodels, masked gradients, and class-level prototypes. The former two are standard in FL pipelines, while prototype sharing aligns with recent prototype-based personalization methods (Liu et al., 2024).

To further examine the sensitivity of prototype sharing, we evaluate the robustness of FRAMP under two forms of prototype perturbation. Gaussian noise and random rotation are commonly used perturbations, and here they serve as stress tests to assess how strongly the model depends on the exact prototype values. Table 4 shows representative results

Table 4: Test accuracy under Gaussian noise and random rotation on CIFAR100.

| | No Noise | GN $a = 0.01$ | GN $a = 0.1$ | GN $a = 0.5$ | Random Rotation |
|---|---|---|---|---|---|
| Local | 42.95 | 42.40 | 41.45 | 41.25 | 41.73 |
| Union | 40.26 | 40.26 | 40.04 | 38.41 | 39.02 |

under strong perturbations (GN $a = 0.5$ and random rotation), where degradation occurs. More results across submodel sizes are in Appendix D.6, which overall show minimal accuracy loss in most cases, suggesting that FRAMP leverages prototypes mainly for semantic guidance rather than precise reconstruction.

Although FRAMP involves uploading masked model updates, its use of personalized submodels and dynamic masking limits the amount of consistent gradient information observed by the server.

These mechanisms reduce cross-round consistency that gradient inversion attacks typically rely on. Moreover, FRAMP remains compatible with standard privacy-enhancing techniques, such as secure aggregation and differential privacy, which can be applied in parallel to provide stronger privacy protection.

**Additional Experiments in Appendix** Appendix D.1 reports results under severe system heterogeneity with five model sizes. Appendix D.2 presents experiments under different levels of data heterogeneity, including stronger heterogeneity ($\alpha = 0.1, 0.05$), milder heterogeneity ($\alpha = 0.5, 0.7$), and the IID setting. Appendix D.3 evaluates scenarios with more participants per round.

## 6 CONCLUSION

We proposed FRAMP, a unified framework for personalized and resource-adaptive FL. By integrating client-aware model generation, adaptive submodel extraction, and semantic alignment, FRAMP enables clients to obtain effective and personalized submodels. Extensive Experiments demonstrate robust personalization and generalization. FRAMP achieves more balanced performance across model-size groups, maintains strong accuracy for clients with limited budgets, and generalizes well to unseen compute budgets and new clients. While our study focuses on magnitude-based importance and hypernetwork-based model generation, exploring alternative importance estimators or more expressive generators are promising directions for future work.

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

# A RELATED WORKS

## A.1 MODEL SPARSIFICATION IN FL

Model sparsification, or pruning, has gained attention following the introduction of the lottery ticket hypothesis (Frankle & Carbin, 2018), which suggests that within large models lie smaller subnetworks that can be trained to perform effectively. Early studies (Liu et al., 2015) leveraged sparsity to reduce computational overhead. Recently, hardware innovations have further advanced the feasibility of training and deploying sparse models (Kurtz et al., 2020; Hoefler et al., 2021; Iofinova et al., 2022).

In FL, two main approaches are used to obtain sparse submodels: dense-to-sparse (Li et al., 2021; Isik et al., 2022; Jiang et al., 2022; Wang et al., 2024), which begins with a full model, and sparse-to-sparse (Li et al., 2020a; Seo et al., 2021; Bibikar et al., 2022; Dai et al., 2022; Mugunthan et al., 2022), which starts with a sparse model. Both methods typically distribute a shared global model to clients, and then prune or refine it locally. However, these techniques struggle under heterogeneous resource constraints, where some clients cannot even load the full model due to limited capacity.

To address this, algorithms have emerged that generate submodels tailored to each client's computational budget. For example, Flado (Liao et al., 2023) allows the server to customize submodels based on client computation budgets by ranking parameters according to importance. While effective, this method requires clients to store and update importance scores for all parameters, introducing non-trivial storage and computation overhead. FIARSE (Wu et al., 2024) avoids this burden by leveraging the observation that parameter magnitude often correlates with importance. It uses parameter values themselves as implicit importance indicators, thereby simplifying the process and reducing overhead on resource-limited clients.

## A.2 HYPERNETWORKS

HNs (Klein et al., 2015) are neural networks designed to generate weights for another network. The target weights can be dynamically adapted based on the HNs' input vectors. (Klocek et al., 2019; Navon et al., 2020). SMASH (Brock et al., 2017) introduced HNs into Neural Architecture Search (NAS) by encoding architectures as 3D tensors using a memory channel mechanism. GHN (Zhang et al., 2018) was proposed for anytime prediction, focusing on both inference efficiency and the speed-accuracy trade-off. In FL settings, pFedHN (Shamsian et al., 2021) employed HNs with task-specific embeddings for personalization, while hFedGHN (Xu et al., 2023) extended GHNs to support heterogeneous local models through graph-based reasoning.

Direct comparison with HN-based FL baselines is not applicable, as they are designed for uniform models and do not support varying model sizes.

# B ALGORITHM

---

**Algorithm 1** FRAMP

**Input:** Communication rounds $R$, number of clients $N$, local training rounds $T$.

**Server:**

1: **for** $r = 1$ to $R$ **do**
2:     Sample a subset of clients $\mathcal{A} \subseteq [N]$.
3:     **for all** $n \in \mathcal{A}$ **do**
4:         Generate customized full model
        $\boldsymbol{\omega}_n^r := H(\mathbf{v}_n; \boldsymbol{\varphi}^r)$.
5:         Compute client-specific mask $\mathcal{M}_n^r(\boldsymbol{\omega}_n^r)$.
6:         Send submodel $\hat{\boldsymbol{\omega}}_n^r = \boldsymbol{\omega}_n^r \odot \mathcal{M}_n^r(\boldsymbol{\omega}_n^r)$
        and global prototypes $\mathbf{P}_g^r$ to client $n$.
7:         CLIENTUPDATE$(n, \hat{\boldsymbol{\omega}}_n^r, \mathbf{P}_g^r)$
8:     **end for**
9:     Update $\boldsymbol{\varphi}^{r+1} = \boldsymbol{\varphi}^r - \beta \left(\nabla_{\boldsymbol{\varphi}^r} \hat{\boldsymbol{\omega}}_n^r\right)^\top \Delta \hat{\boldsymbol{\omega}}_n^r$.
10:    Update $\mathbf{P}_g^{r+1} \leftarrow \{\mathbf{P}_n^{r,c}\}_{c=1}^C$.
11: **end for**

**ClientUpdate** $(n, \hat{\boldsymbol{\omega}}_n^r, \mathbf{P}_g^r)$:

1: Set $\hat{\boldsymbol{\omega}}_n^{r,1} = \hat{\boldsymbol{\omega}}_n^r$
2: **for** $t = 1$ to $T$ **do**
3:     Update parameters
    $\hat{\boldsymbol{\omega}}_n^{r,t+1} = \hat{\boldsymbol{\omega}}_n^{r,t} - \alpha \nabla_{\hat{\boldsymbol{\omega}}_n^{r,t}} \mathcal{L}_n(\hat{\boldsymbol{\omega}}_n^{r,t})$.
4:     Update local prototypes $\{\mathbf{P}_n^{r,c}\}_{c=1}^C$.
5: **end for**
6: Compute update: $\Delta \hat{\boldsymbol{\omega}}_n^r = \hat{\boldsymbol{\omega}}_n^{r,T} - \hat{\boldsymbol{\omega}}_n^{r,1}$.
7: **return** $\Delta \hat{\boldsymbol{\omega}}_n^r, \{\mathbf{P}_n^{r,c}\}_{c=1}^C$

---

## C    EXPERIMENTAL SETUPS

### C.1    DATASETS

For CIFAR10 and CIFAR100, the datasets contain 50K training samples and 10K test samples, which are then divided among 100 clients using a Dirichlet distribution with concentration parameter $\alpha = 0.3$. For ogbn-arxiv, 40% of nodes are used for training, 30% for validation, and the remaining 30% for testing. To distribute the graph among clients, we employ the METIS graph partitioning algorithm (Karypis, 1997), which segments the original graph into a specified number of disjoint subgraphs. Each client is assigned one of these non-overlapping subgraphs. In our case, setting METIS to 100 yields 100 disjoint subgraphs, each corresponding to a client in the federated setting.

### C.2    HYPERPARAMETERS

Table 5 lists the hyperparameters used in the experiments.

Table 5: Hyperparameter Settings

|  | CIFAR10 | CIFAR100 | ogbn-arxiv |
|---|---|---|---|
| Local Epochs | 100 | 100 | 120 |
| Batch Size | 32 | 32 | - |
| Communication Rounds | 800 | 800 | 200 |
| Learning rate | {0.1, 0.5} | {0.1, 0.4} | {0.1} |
| HN Learning rate | 0.12 | 0.08 | 0.1 |
| $\lambda$ | 0.7 | 0.7 | 0.2 |

## D    MORE EXPERIMENTS

### D.1    SYSTEM HETEROGENEITY

**System heterogeneity with five model sizes**    We evaluate a setting with five model capacities, $\gamma' = 0.04, 0.16, 0.36, 0.64, 1.0$, with clients evenly assigned to each level. This experiment illustrates the flexibility of our approach, which can naturally extend to different numbers of capacity levels or alternative client distributions.

Table 6: Test accuracy of submodels across five submodel sizes.

| Method | CIFAR-100 | | | | | | |
|---|---|---|---|---|---|---|---|
|  | Local | 0.04 | 0.16 | 0.36 | 0.64 | 1.0 | Union |
| HeteroFL | 35.01 | 30.15 | 33.60 | 36.10 | 37.75 | 37.45 | 32.60 |
| FedRolex | 38.51 | 28.40 | 37.75 | 41.10 | 42.10 | 43.19 | 36.09 |
| ScaleFL | 42.43 | 42.90 | 44.75 | 42.49 | 42.05 | 39.95 | 40.21 |
| FIARSE | 45.94 | 44.35 | 45.65 | 47.80 | 47.00 | 44.90 | 42.61 |
| FRAMP | **46.71** | **46.75** | **46.65** | **47.85** | **47.35** | 44.95 | **43.67** |

### D.2    DATA HETEROGENEITY

To examine the effect of stronger data heterogeneity, we partition CIFAR100 into 100 clients using a Dirichlet distribution with concentration parameter $\alpha = 0.1$ and $\alpha = 0.05$. The results are reported in Table 8 and Table 7. Compared with the default $\alpha = 0.3$, this setting yields a more skewed label distribution across clients, providing a stricter test of robustness under highly non-IID conditions.

To explore the impact of less heterogeneous setting, we conduct experiments with $\alpha = 0.5$ and $\alpha = 0.7$, where client data distributions become more balanced. The corresponding results are presented in Table 9 and Table 10.

In addition, we include experiments under the IID partition with results shown in Table 11.

Table 7: Test accuracy on CIFAR100 with stronger data heterogeneity ($\alpha = 0.05$).

| Method | CIFAR100 | | | | | |
|---|---|---|---|---|---|---|
| | Local | 1/64 | 1/16 | 1/4 | 1.0 | Union |
| HeteroFL | 22.24 | 17.28 | 21.96 | 22.16 | 27.56 | 21.48 |
| FedRolex | 12.63 | 3.08 | 5.84 | 16.12 | 25.48 | 11.67 |
| ScaleFL | 29.14 | 28.68 | 32.04 | 28.64 | 27.20 | 27.71 |
| FIARSE | 29.29 | 25.68 | 31.84 | **29.24** | 30.40 | 29.25 |
| FRAMP | **30.92** | **31.32** | **33.12** | 28.72 | **30.52** | **30.10** |

Table 8: Test accuracy on CIFAR100 with stronger data heterogeneity ($\alpha = 0.1$).

| Method | CIFAR100 | | | | | |
|---|---|---|---|---|---|---|
| | Local | 1/64 | 1/16 | 1/4 | 1.0 | Union |
| HeteroFL | 25.80 | 20.68 | 23.92 | 29.36 | 29.24 | 23.79 |
| FedRolex | 16.06 | 2.56 | 8.12 | 23.32 | 30.24 | 15.04 |
| ScaleFL | 33.74 | 31.36 | 35.08 | 37.76 | 30.76 | 31.61 |
| FIARSE | 34.64 | 29.92 | 36.04 | **40.08** | 32.52 | 34.10 |
| FRAMP | **36.57** | **36.56** | **38.16** | 38.96 | **32.60** | **35.40** |

Table 9: Test accuracy on CIFAR100 with milder data heterogeneity ($\alpha = 0.5$).

| Method | CIFAR100 | | | | | |
|---|---|---|---|---|---|---|
| | Local | 1/64 | 1/16 | 1/4 | 1.0 | Union |
| HeteroFL | 30.73 | 25.64 | 28.36 | 33.16 | 35.76 | 29.18 |
| FedRolex | 19.06 | 3.28 | 10.52 | 27.28 | 35.16 | 18.81 |
| ScaleFL | 40.64 | 35.48 | 43.16 | 43.20 | 40.72 | 37.30 |
| FIARSE | 41.13 | 37.88 | 43.08 | 43.80 | 39.76 | 38.91 |
| FRAMP | **42.22** | **40.96** | **43.20** | **43.88** | **40.84** | **40.17** |

Table 10: Test accuracy on CIFAR100 with milder data heterogeneity ($\alpha = 0.7$).

| Method | CIFAR100 | | | | | |
|---|---|---|---|---|---|---|
| | Local | 1/64 | 1/16 | 1/4 | 1.0 | Union |
| HeteroFL | 31.16 | 28.60 | 28.80 | 31.40 | 35.84 | 29.69 |
| FedRolex | 19.06 | 3.12 | 11.76 | 26.80 | 34.56 | 18.80 |
| ScaleFL | 42.05 | 39.96 | 43.28 | **43.44** | 41.52 | 38.44 |
| FIARSE | 40.31 | 38.20 | 41.44 | 41.24 | 40.36 | 37.71 |
| FRAMP | **43.25** | **44.84** | **44.32** | 42.72 | 41.12 | **40.53** |

Table 11: Test accuracy on CIFAR100 with IID split.

| Method | CIFAR100 | | | | | |
|---|---|---|---|---|---|---|
| | Local | 1/64 | 1/16 | 1/4 | 1.0 | Union |
| HeteroFL | 31.87 | 26.88 | 31.40 | 33.36 | 35.84 | 29.11 |
| FedRolex | 20.55 | 2.68 | 12.24 | 30.16 | 37.12 | 19.31 |
| ScaleFL | 44.37 | 41.80 | 44.76 | **46.52** | 44.40 | 41.15 |
| FIARSE | 42.38 | 40.12 | 43.56 | 44.52 | 41.32 | 40.63 |
| FRAMP | **46.20** | **45.88** | **48.32** | 45.84 | **44.76** | **43.06** |

### D.3 MORE PARTICIPATE CLIENTS

**Participation rate of 20%** Table 12 reports test accuracy on CIFAR100 when the client participation ratio is set to 20%. For FRAMP, the gains are less pronounced for clients with large submodels, but it delivers more balanced performance overall, particularly benefiting clients with limited resources, its primary design objective.

Table 12: Test accuracy of submodels across four submodel sizes with 20% participation rate.

| Method | CIFAR100 | | | | | |
|--------|-------|-------|-------|-------|-------|-------|
| | Local | 1/64 | 1/16 | 1/4 | 1.0 | Union |
| HeteroFL | 32.23 | 28.32 | 31.52 | 33.96 | 35.12 | 30.30 |
| FedRolex | 33.00 | 21.36 | 34.12 | 36.72 | 39.80 | 31.33 |
| ScaleFL | 39.57 | 37.92 | 39.60 | 41.84 | 38.92 | 37.63 |
| FIARSE | 42.27 | 40.32 | 43.28 | **43.52** | **41.96** | 38.97 |
| FRAMP | **42.72** | **41.80** | **44.80** | 43.28 | 41.00 | **39.59** |

### D.4 SENSITIVITY OF $\lambda$

To examine the sensitivity of FRAMP to the alignment loss weight $\lambda$, we conduct experiments on CIFAR10 with $\lambda$ in $0.1, 0.2, \ldots, 0.9$. For each setting, models are evaluated across submodel sizes $\gamma' \in 1/64, 1/16, 1/4, 1.0$, along with overall averages for Local and Union test set performance.

The results in Fig. 7 show that FRAMP maintains stable performance for a broad range of $\lambda$. Moderate values, e.g., $\lambda = 0.5$–$0.7$, yield the best overall accuracy. Very small ($\lambda = 0.1$) or large ($\lambda = 0.9$) values lead to reduced accuracy, reflecting under- or overemphasis on the alignment term.

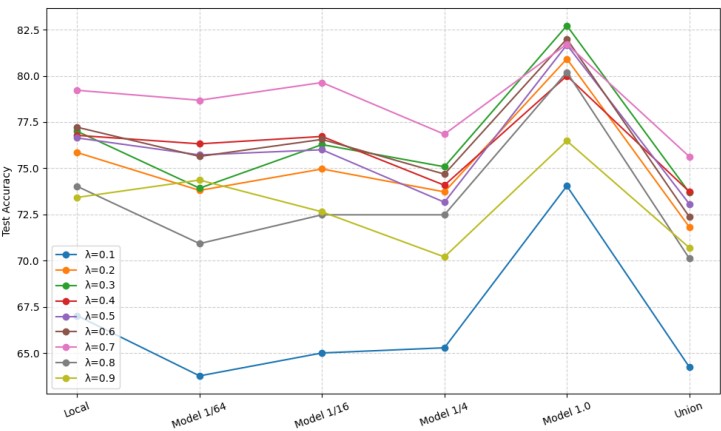

Figure 7: Test accuracy across different model sizes (Local, 1/64, 1/16, 1/4, 1.0, Union) under varying values of the alignment weight $\lambda$.

### D.5 COMPUTATIONAL AND COMMUNICATION OVERHEAD DISCUSSION

In FRAMP, the additional communication overhead compared to existing heterogeneous FL baselines is:

- **Client descriptor.** Each client uploads its descriptor once during initialization. A 128-dimensional descriptor is about 0.5 KB.

- **Class prototypes.** Class prototypes are exchanged each communication round. For ResNet18 on CIFAR10, a full set of prototypes (10 classes × 512 dimensions) is about 20 KB.

**Scale with model size.** Model parameters can be produced in a layer-wise or block-wise manner using low-dimensional latent kernels that are reshaped, sliced, or concatenated to obtain the final parameters (Zhang et al., 2018). This design enables the HNs to support large-scale models.

**Scale with number of clients.** In each communication round, the server generates full models only for the participating clients. These models can be produced sequentially, keeping the peak server memory footprint effectively stable regardless of the total number of clients. Storing client descriptors requires only lightweight data and has a negligible impact on memory consumption.

### D.6 PRIVACY DISCUSSION

Specifically, we apply two types of noise: 1) Gaussian noise. Each prototype is perturbed as $\tilde{\mathbf{P}}_n^c = \mathbf{P}_n^c + \epsilon$, where $\epsilon \sim \mathcal{N}(0, \sigma^2 I)$ and $\sigma = a \cdot \|\mathbf{P}_n^c\|$. 2) Random rotation. Each prototype is transformed by a random orthogonal matrix, i.e., $\tilde{\mathbf{P}}_n^c = \mathbf{Q}\mathbf{P}_n^c$ where $\mathbf{Q}^\mathsf{T}\mathbf{Q} = I$. As shown in Table 13, both noise injection strategies result in minimal accuracy degradation. This suggests that FRAMP leverages prototypes for semantic guidance rather than precise reconstruction and is robust to moderate obfuscation.

Table 13: Test accuracy under prototype perturbations on CIFAR100.

| Method | CIFAR100 | | | | | |
|---|---|---|---|---|---|---|
| | Local | 1/64 | 1/16 | 1/4 | 1.0 | Union |
| NoNoise | 42.95 | 43.00 | 44.08 | 43.72 | 41.00 | 40.26 |
| GN $a = 0.01$ | 42.40 | 43.36 | 44.64 | 41.80 | 39.80 | 40.26 |
| GN $a = 0.1$ | 41.45 | 43.20 | 43.04 | 41.12 | 38.44 | 40.04 |
| GN $a = 0.5$ | 41.25 | 43.04 | 43.32 | 41.00 | 37.64 | 38.41 |
| Random Rotation | 41.73 | 43.52 | 44.24 | 40.84 | 38.32 | 39.02 |

### D.7 DISCUSSION OF SECTION 3.2

To generate Fig. 1, for each model-size group (1.0, 1/4, 1/16, 1/64), we collect the masks produced by clients in that group and compute the average activation probability for each parameter. We then plot the cumulative coverage, which shows the cumulative proportion of activated parameters covered up to a given parameter index. The curve reflects how often each parameter is selected by clients within the same model-size group.

If masks from different clients are diverse, each parameter should have a similar chance of being selected. In this case, the curve is close to a straight diagonal line, as in the size = 1.0 case. If many clients select overlapping subsets of parameters, the curve rises quickly in regions where parameters are frequently selected and becomes nearly flat in regions where parameters are rarely used. This pattern clearly appears in the size = 1/16 and 1/64 groups.

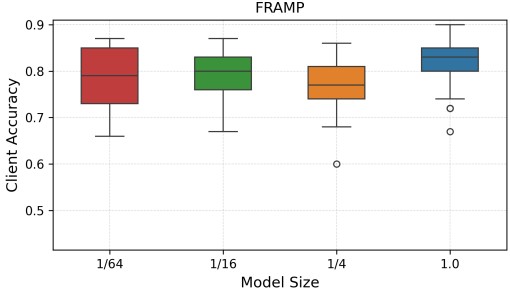

Figure 8: Client accuracy distribution of submodels across different sizes. Each box represents the variation in accuracy among clients.

# E  LLM USAGE

We acknowledge the use of large language models (LLMs) as assistive tools in preparing this manuscript. LLMs were employed solely for polishing the writing and refining grammar, improving clarity, and enhancing fluency. LLMs were **NOT** used to generate research ideas, conduct analysis, or produce results. All conceptual contributions, theoretical developments, experimental designs, and interpretations are the authors' own.

