# OpenReview forum: "Representation-Aligned Multi-Scale Personalization for Federated Learning"
_ICLR.cc/2026/Conference — Submitted to ICLR 2026_

### Official Review · Reviewer_FCqt · 2025-10-15

**Soundness:** 4
**Presentation:** 4
**Contribution:** 4
**Rating:** 8
**Confidence:** 4

**Summary:**

In federated learning, accommodating clients with diverse computational resources remains challenging. Existing approaches often derive client submodels from a shared global backbone, which restricts structural diversity and limits representational adaptation. FRAMP addresses this by generating client-specific models from compact descriptors, enabling personalization to both data and resource constraints. Each client trains a lightweight tailored submodel while aligning its representations to preserve global semantic consistency.

**Strengths:**

1.	**Elegant Solution to a Practical Problem**: The proposed approach demonstrates a clean architectural design: it integrates resource-awareness and personalization within a unified descriptor-driven framework, avoiding additional communication or coordination complexity. Resource heterogeneity is a critical bottleneck for real-world FL deployment, particularly in cross-device settings. The paper's motivation and methodological framing are both relevant to industry-scale scenarios (e.g., mobile or edge-device training).

2.	**Dynamic and Adaptive Submodel Extraction**: The paper’s adaptive submodel extraction mechanism (where parameter importance is dynamically encoded during training) is a notable contribution. It allows submodels to evolve organically based on training dynamics, rather than relying on static pruning or masking heuristics. This reduces manual tuning and ensures that model capacity allocation remains responsive to the data distribution and client conditions.

3. **Strong Writing and Technical Clarity**: The exposition is clear, with well-structured sections that balance algorithmic intuition and formal derivation. The authors successfully convey the relationship between descriptor generation, submodel extraction, and alignment.

**Weaknesses:**

1.	**Unclear Motivation Behind "Underutilized Parameters" Argument**:

The paper states:
"Although FIARSE supports dynamic submodel sampling in each round, parameters with low importance scores are seldom chosen, resulting in limited training opportunities for these parameters. Consequently, a significant portion of the model remains underutilized, leading to high structural similarity and reduced diversity among submodels."

First, it is unclear why low-importance parameters should be trained if they consistently exhibit marginal contribution to model performance.

Second, diversity among submodels is presented as an implicit goal, but the paper does not provide a theoretical or empirical argument explaining why greater structural diversity necessarily improves generalization or fairness.

2.	**Fixed Client Descriptors May Limit Adaptivity**:

At initialization, each client transmits its descriptor to the server once, which remains fixed throughout training. This design choice could hinder adaptivity when client conditions evolve, for example, due to fluctuating compute availability, memory constraints, or battery limitations. A dynamic descriptor that updates periodically or conditionally could better reflect real-world device heterogeneity. The paper would benefit from either (a) empirical evidence that descriptors remain valid throughout training, or (b) discussion of mechanisms for re-synchronization.

**Questions:**

1.	How is a client's computational capacity (which indicates how large of a model can be assigned to it for training) computed in real-life?

2.	Does the client receive both the submodel parameters and the corresponding mask defining trainable weights? If so, does the mask transmission effectively double the communication cost? A quantitative comparison of FRAMP’s communication and computation overhead against baselines would help substantiate the efficiency claims.

3.	The paper mentions retaining only parameters above an adaptive importance threshold. Conceptually, what does this threshold represent; magnitude, gradient sensitivity, or representational salience? Why should retaining only “high-value” parameters always yield optimal performance? It is conceivable that a complementary mix of high- and low-importance parameters could yield better local minima or smoother gradient flow. Additionally, pruning aggressively might cause gradient instability or blow-up, especially under non-IID conditions.

4.	What is the difference between local prototype and vector used for the personalized model creation?

---

> ### Author Response · Authors · 2025-11-23
> **1/2 - Rebuttal by Authors**
>
> We thank the reviewer for the encouraging evaluation and the insightful comments. We respond to your questions below.
>
> > W1: First, it is unclear why low-importance parameters should be trained if they consistently exhibit marginal contribution to model performance. Second, diversity among submodels is presented as an implicit goal, but the paper does not provide a theoretical or empirical argument explaining why greater structural diversity necessarily improves generalization or fairness.
>
> Thank you for the comment. Our intention is not to argue that parameters ranked as low-importance under FIARSE’s global score are inherently valuable or should be trained. Rather, these parameters become low-ranked because a single global importance score is shared across all clients. This ranking reflects the dominant data distributions, causing many lower-ranked parameters to receive little training opportunity, even if they could be useful for some clients. Under **heterogeneous distributions**, different clients may benefit from different subsets of parameters, and a shared global ranking suppresses parameters that would be useful for some clients but not for others. This restricts each client’s ability to obtain a submodel that fits its own data, which is especially detrimental for clients with small compute budgets.
>
> Regarding structural diversity, we clarify that diversity is not an objective by itself. In FIARSE, clients within the same model-size group activate highly overlapping subsets of parameters, resulting in similar submodel structures **even though the clients have heterogeneous data distributions**. This lack of diversity is a symptom of insufficient personalization under data and system heterogeneity. Clients with distinct data distributions and model sizes may benefit from different parameter subsets, but the shared global ranking forces them to extract submodels from nearly identical regions of the parameters, leading to performance imbalance. FRAMP addresses this by generating personalized full models, allowing submodels to naturally differ. The resulting structural diversity emerges as a consequence of personalization and contributes to more balanced performance across heterogeneous clients, rather than serving as an explicit objective.
>
> > W2: The paper would benefit from either (a) empirical evidence that descriptors remain valid throughout training, or (b) discussion of mechanisms for re-synchronization.
>
> Thank you for the insightful comment. In our study, we explore this question empirically. Table 2 compares several descriptor strategies, including (i) fixed descriptors, (ii) one-hot descriptors, and (iii) periodically updated descriptors. The results show that periodic updates do not yield consistent gains over the fixed design in our setting.
>
> At the same time, FRAMP does not assume that descriptors must remain fixed. Clients can update their descriptors whenever their local conditions change, and the framework can naturally accommodate such updates without modifying its core design. Exploring dynamic, evolving condition scenarios is an interesting direction for future work.
>
> > Q1: How is a client's computational capacity (which indicates how large of a model can be assigned to it for training) computed in real-life?
>
> Thank you for the question. The computational capacity is determined on the client side and provided to the server. Clients can estimate their capacity based on measurable properties such as available memory, device FLOPs, battery or energy constraints, and acceptable training or inference latency. Communication limits may also be considered, as edge devices often have lower upload bandwidth than download.
> For example, consider an edge device such as a Raspberry Pi 1 Model A, which has 256 MB of memory. Training ResNet18 with a small batch of 8 requires over 560 MB [1], which exceeds its capability. A device in this situation would request a smaller submodel. The capacity of the device can be estimated via its memory size or determined via testing [2].
>
> [1] FedDSE: Distribution-aware Sub-model Extraction for Federated Learning over Resource-constrained Devices, WWW 2024.
>
> [2] https://developer.nvidia.com/blog/gpu-memory-essentials-for-ai-performance/

---

> ### Author Response · Authors · 2025-11-23
> **2/2 - Rebuttal by Authors**
>
> > Q2: Does the client receive both the submodel parameters and the corresponding mask defining trainable weights? If so, does the mask transmission effectively double the communication cost? A quantitative comparison of FRAMP’s communication and computation overhead against baselines would help substantiate the efficiency claims.
>
> Thank you for the question. We clarify that FRAMP does not transmit masks to clients. Submodel extraction is performed on the server, and clients receive only the submodel, consistent with existing heterogeneous FL baselines such as FIARSE. Under the same model-size budget, FRAMP transmits the same amount of parameters as these baselines. The additional transmitted descriptor is about 0.5KB (upload once during initialization), and class prototypes are about 20KB. We have included additional information in Appendix D.5.
>
> > Q3: Conceptually, what does this threshold represent; magnitude, gradient sensitivity, or representational salience? Why should retaining only “high-value” parameters always yield optimal performance? It is conceivable that a complementary mix of high- and low-importance parameters could yield better local minima or smoother gradient flow. Additionally, pruning aggressively might cause gradient instability or blow-up, especially under non-IID conditions.
>
> Thank you for the insightful question. The adaptive threshold in FRAMP is a magnitude-based cutoff that selects the Top-K parameters required by each client’s model size budget. The use of magnitude-based Top-K does not assume that magnitude is a perfect measure of importance. Instead, it follows extensive pruning literature [3,4,5] that has shown weight magnitude to be an inexpensive and effective proxy for parameter importance.
>
> In FRAMP, magnitude-based ranking is particularly appropriate because all clients share a single hypernetwork. The hypernetwork learns a structured parameterization that naturally emphasizes influential directions in the weight space, which often appear as larger-magnitude parameters. Consequently, Top-K extracted submodels preserve these informative components, and our experiments show that even very small submodels retain strong accuracy under this strategy. Although more refined importance estimators that incorporate additional factors exist, they typically require second-order information or additional passes.
>
> Regarding gradient stability, FRAMP does not perform iterative pruning during optimization. Each client trains on a fixed submodel structure throughout a communication round, and the hypernetwork parameters evolve smoothly across rounds. This avoids the gradient-disruption issues commonly associated with pruning-based training.
>
> [3] Parameter Efficient Training of Deep Convolutional Neural Networks by Dynamic Sparse Reparameterization, ICML 2019.
>
> [4] Top-kast: Top-k Always Sparse Training, NIPS 2020.
>
> [5] FIARSE: Model-heterogeneous Federated Learning via Importance-aware Submodel Extraction, NIPS 2024.
>
> > Q4: What is the difference between local prototype and vector used for the personalized model creation?
>
> Thank you for the question. Client descriptors and local prototypes serve different purposes and are constructed in entirely different ways.
>
> The descriptor is obtained by averaging the encoded embeddings produced by a feature extractor over the client’s local dataset. It captures coarse, client-level information about data characteristics and is used solely as the input to the hypernetwork for generating a personalized full-size model. It contains no class-level semantic information and remains fixed during training.
>
> Local prototypes are class-specific representations computed during training based on the embeddings produced by the local model. They are updated every communication round and are used to align class-level semantics across clients.
>
> In summary, the descriptors guide personalized model generation at the client level, while local prototypes support class-level representation alignment during training.

---

> > ### Comment · Reviewer_FCqt · 2025-11-25
> >
> > Thank you for the detailed response. I will keep my current score.

---

> > > ### Author Response · Authors · 2025-11-25
> > >
> > > We sincerely appreciate your continued support. Thank you again for your thoughtful comments.

---

### Official Review · Reviewer_SYMr · 2025-10-27

**Soundness:** 3
**Presentation:** 3
**Contribution:** 2
**Rating:** 4
**Confidence:** 4

**Summary:**

The paper aim to tackle the limitation of structural diversity and representational adaptation focusing on the challenge of integrating clients with diverse resource constraints. The paper proposes FRAMP (Federated Representation-Aligned Multi-scale Personalization), a framework for model-heterogeneous, personalized federated learning (FL). FRAMP (i) uses a server-side hypernetwork to generate a client-specific full model from a compact per-client descriptor; (ii) performs adaptive submodel extraction via global Top-K magnitude masking to satisfy each client’s sparsity/computation budget; and (iii) adds prototype-guided representation alignment, where clients upload class prototypes and align to server-aggregated global prototypes without any shared public dataset. Experiments of image classification and node classification on graph-structured data show the personalization, generalization, robustness to unseen budgets and clients compare to the submodel extraction approaches.

**Strengths:**

- FRAMP addresses the limitations of relying on a single shared global backbone by proposing a client-aware model generation mechanism. This mechanism leverages compact client descriptors to instantiate personalized full-size models, allowing for fine-grained adaptation to both the client's data characteristics and their computational budgets. Building upon this, FRAMP uses an adaptive submodel extraction strategy that dynamically selects tailored, sparse submodels based on evolving parameter importance, ensuring resource constraints are met without introducing additional overhead.

- Extensive experiments on various benchmarks demonstrate that FRAMP consistently outperforms most strong baselines on the reported setups that support model heterogeneity. FRAMP maintains strong adaptability and comparable accuracy when generalizing to unseen client scenarios or missing capacity profiles.

**Weaknesses:**

- The server must generate full client-specific weights, apply global Top-K masking, and backpropagate through the hypernetwork using updates. The paper reports server time per round (slightly higher in round 1, similar afterward), but a deeper accounting of compute/memory (e.g., peak RAM, scaling with client/model size, HN size) is missing. In large-scale vision backbones, full weight materialization per participating client can be non-trivial.

- FRAMP still exposes an attack surface for DLG/gradient-inversion–style attacks, since each round transmits masked model updates  and class prototypes to the server, and clients upload a descriptor at initialization. The paper argues that per-client personalized submodels and dynamic Top-K masking disrupt cross-round consistency and thus make such attacks harder, but this is a qualitative claim without formal guarantees. The presented “privacy analysis” mainly tests accuracy robustness to noisy/rotated prototypes rather than resistance to data reconstruction or label-distribution leakage. Without additional mechanisms such as secure aggregation and differential privacy, the risk of sensitive information leakage cannot be conclusively ruled out.

- I know that the space is limited due to page limitations, but the Conclusion section does not fully contain the contributions/limitation/future works of the paper.

**Questions:**

- When calculating TopK, the authors only keep parameters with large absolute values, but what about cases where a specific parameter value is small, yet it could play a significant role when considering its combination (interaction) with other parameters?

- Please include experiments under IID client partitions to disentangle the benefits of FRAMP from non-IID effects.

---

> ### Author Response · Authors · 2025-11-23
> **1/2 - Rebuttal by Authors**
>
> We thank the reviewer for the detailed questions. We address your concerns below.
>
> > W1: The server must generate full client-specific weights, apply global Top-K masking, and backpropagate through the hypernetwork using updates. The paper reports server time per round (slightly higher in round 1, similar afterward), but a deeper accounting of compute/memory (e.g., peak RAM, scaling with client/model size, HN size) is missing. In large-scale vision backbones, full weight materialization per participating client can be non-trivial.
>
> Thank you for the question. This concern relates to the feasibility of using hypernetworks to generate large models. We clarify this below.
>
> 1. **Hypernetworks can scale to large models in practice.**
> Although the hypernetwork has a fixed output size, it can still generate weights for layers of different shapes by concatenating multiple same size kernels or by slicing in spatial dimensions [2]. This enables large models to be produced in a layer-wise or block-wise manner. Several works have successfully generated ResNet-scale models using hypernetworks [1,3], demonstrating the practical feasibility.
>
> 2. **Server-heavy, client-light design is reasonable.**
> FL typically assumes that the server has substantially greater compute and memory resources than edge devices. In FRAMP, the hypernetwork is executed entirely on the server, while the client-side training procedure remains identical to that of standard heterogeneous FL methods. Although the hypernetwork introduces additional server-side memory usage, the per-round training time in our implementation remains comparable.
>
>
> 3. **Server compute grows modestly and is not tied to the total number of clients.**
> In each communication round, the server generates full models only for the participating clients. When more clients participate, models can be generated sequentially, so the memory footprint does not scale with client size. Increasing the number of clients primarily increases only lightweight data, such as stored descriptors.
>
> We have added a short discussion of server-side scaling behavior in Appendix D.5.
>
> [1] Hypernetworks, ICLR 2017.
>
> [2] Graph Hypernetworks for Neural Architecture Search, ICLR 2019.
>
> [3] A Parameter Prediction for Unseen Deep Architectures, NeurIPS 2021.
>
> >W2: FRAMP still exposes an attack surface for DLG/gradient-inversion–style attacks. The paper argues that per-client personalized submodels and dynamic Top-K masking disrupt cross-round consistency and thus make such attacks harder, but this is a qualitative claim without formal guarantees. The presented “privacy analysis” mainly tests accuracy robustness to noisy/rotated prototypes rather than resistance to data reconstruction or label-distribution leakage. Without additional mechanisms such as secure aggregation and differential privacy, the risk of sensitive information leakage cannot be conclusively ruled out.
>
> Thank you for the detailed comment. We clarify that our prototype perturbation experiment is not intended to simulate gradient inversion attacks or to claim privacy guarantees. The purpose of this experiment is to evaluate the sensitivity of the additional information that FRAMP transmits, which is the class prototypes. Gaussian noise and random rotation are commonly used perturbation defenses for prototype-based representations, and our experiment examines whether FRAMP remains robust when such perturbations are applied. This robustness test is not a substitute for resistance to data reconstruction, and we have revised the wording to avoid suggesting such an interpretation.
>
> We agree that robustness under prototype perturbations does not constitute a full privacy analysis, and FRAMP alone cannot eliminate the possibility of data or label distribution leakage. Stronger privacy protection would require additional mechanisms such as secure aggregation or differential privacy, which can be incorporated into the FL framework. These techniques are fully compatible with FRAMP and can be incorporated without affecting the framework’s core design. This is the same as for existing FL methods that do not include formal privacy defenses, see [4,5].
>
> We also note that the risk of reconstruction attacks applies similarly to existing baselines, which transmit model updates without formal privacy defenses. For **fairness of comparison**, we did not augment any method, including FRAMP and all baselines, with additional privacy mechanisms. FRAMP would benefit equally from such protections if they are applied.
>
> We have updated the manuscript to more clearly position our study.
>
> [4] Personalized Federated Learning Using Hypernetworks, ICML 2021.
>
> [5] Subgraph Federated Learning for Local Generalization, ICLR 2025.
>
> > W3: the Conclusion section does not fully contain the contributions/limitation/future works of the paper.
>
> Thank you for the comment. We have updated the conclusion accordingly.

---

> ### Author Response · Authors · 2025-11-23
> **2/2 - Rebuttal by Authors**
>
> > Q1: When calculating TopK, the authors only keep parameters with large absolute values, but what about cases where a specific parameter value is small, yet it could play a significant role when considering its combination (interaction) with other parameters?
>
> Thank you for the insightful question. We agree that a parameter with a small magnitude may still contribute through interactions with other parameters. Our use of magnitude-based Top-K does not assume that magnitude is a perfect measure of importance. Instead, it follows extensive evidence from the model pruning literature, showing that weight magnitude provides an inexpensive and effective proxy for parameter importance. Prior studies [6,7,8] have consistently shown that magnitude-based selection performs competitively with more expensive criteria.
>
> In FRAMP, magnitude-based ranking is particularly well suited because all clients share a single hypernetwork. The hypernetwork learns a structured parameterization that naturally emphasizes influential directions in the weight space, which often appear as larger-magnitude parameters. Consequently, Top-K extracted submodels preserve these informative components, and our experiments show that even very small submodels retain strong accuracy under this strategy.
>
> [6] Parameter Efficient Training of Deep Convolutional Neural Networks by Dynamic Sparse Reparameterization, ICML 2019.
>
> [7] Top-kast: Top-k Always Sparse Training, NIPS 2020.
>
> [8] FIARSE: Model-heterogeneous Federated Learning via Importance-aware Submodel Extraction, NIPS 2024.
>
> > Q2: Please include experiments under IID client partitions to disentangle the benefits of FRAMP from non-IID effects.
>
> Thank you for the suggestion. The results in Table 4 (Also included in Appendix D.2 of the updated manuscript) show that FRAMP achieves strong performance across model-size groups under IID partition setting.
>
>
> **Table 4: Test accuracy on CIFAR100 with IID split.**
>
> | **Method**   | **Local** |  **1/64** |  **1/16** |  **1/4**  |  **1.0**  | **Union** |
> | ------------ | :-------: | :-------: | :-------: | :-------: | :-------: | :-------: |
> | **HeteroFL** |   31.87   |   26.88   |   31.40   |   33.36   |   35.84   |   29.11   |
> | **FedRolex** |   20.55   |    2.68   |   12.24   |   30.16   |   37.12   |   19.31   |
> | **ScaleFL**  |   44.37   |   41.80   |   44.76   | **46.52** |   44.40   |   41.15   |
> | **FIARSE**   |   42.38   |   40.12   |   43.56   |   44.52   |   41.32   |   40.63   |
> | **FRAMP**    | **46.20** | **45.88** | **48.32** |   45.84   | **44.76** | **43.06** |

---

> ### Comment · Reviewer_SYMr · 2025-11-24
>
> The authors appropriately resolved my concerns. I raise the score to 6. In particular, it seems that the parts that can be misunderstood in relation to the privacy discussion have been well corrected. In addition, authors provided appropriate additional experimental results. I will consider whether to increase the score by reviewing other reviews.

---

> > ### Author Response · Authors · 2025-11-24
> >
> > We appreciate your careful consideration and are glad that your concerns have been resolved. Thank you again for the constructive and supportive feedback.

---

### Official Review · Reviewer_MSoF · 2025-10-31

**Soundness:** 3
**Presentation:** 3
**Contribution:** 3
**Rating:** 6
**Confidence:** 3

**Summary:**

This paper proposes FRAMP (Federated Representation-Aligned Multi-Scale Personalization), a unified framework for personalized and resource-adaptive federated learning (FL). It targets the problem of client heterogeneity, both in computational capacity and data distribution, by avoiding reliance on a single global model. FRAMP uses a hypernetwork-based client-aware model generator to produce personalized full-size models from compact client descriptors, an adaptive submodel extraction mechanism based on parameter magnitudes, and a prototype-guided representation alignment module to maintain semantic consistency across clients. Experiments on CIFAR-10, CIFAR-100, and ogbn-arxiv show that FRAMP outperforms state-of-the-art baselines such as HeteroFL, FedRolex, ScaleFL, and FIARSE, especially under extreme heterogeneity, and generalizes well to unseen clients and model sizes.

**Strengths:**

++ Proposes a conceptually unified and technically elegant framework that integrates personalization, sparsity, and semantic alignment.

++ Addresses data and system heterogeneity simultaneously.

++ Provides comprehensive experiments with comparisons across multiple datasets and varying heterogeneity levels, including unseen clients and capacity profiles.

++ Clear writing, well-structured methodology, and solid empirical improvements demonstrate intense engineering rigor and reproducibility potential.

**Weaknesses:**

-- The contribution of the submission stems from providing a unified solution to both system and data-level heterogeneity. However, the techniques to achieve this are all from existing studies, i.e., model generation [1], sub-model extraction [2], and prototype-based alignment [3]. A lack of exploration on the unique challenges of combining these techniques makes the contribution incremental.



-- The submission claims that previous methods all use one fully shared global model as the backbone, and such a way would underutilize some parameters. However, this leads to several concerns:





Do high structural similarity and reduced diversity among submodels bring any crucial limitations? Any evidence?



Figure 2 only demonstrates that a smaller model would obtain lower performance. To support the claim that more activations on the early portion of model parameters, the submission should fix the total model size and only change the activation locations (e.g., early portion → mid portion → late portion) rather than change the model size and activation locations simultaneously.



-- The submission only provides the experimental results of the computation overhead of the hypernetwork in the server. Better to include more results of the overhead on additional data transmissions.



-- Though the Appendix provides the experimental results with Dirichlet-\alpha = 0.1, it is better to include more non-IID configurations.


[1] Personalized Federated Learning using Hypernetworks, ICML 2021.

[2] Fiarse: Model-heterogeneous federated learning via importance-aware submodel extraction, NIPS 2024.

[3] FedProto: Federated Prototype Learning across Heterogeneous Clients, AAAI 2022.

**Questions:**

Please see Weaknesses

---

> ### Author Response · Authors · 2025-11-22
> **1/2 - Rebuttal by Authors**
>
> We appreciate your recognition of the strengths of our work and your helpful suggestions. We respond to your concerns and suggestions as follows.
>
> > W1:  A lack of exploration on the unique challenges of combining these techniques makes the contribution incremental.
>
> Thank you for the comment. FRAMP considers a federated setting where both system and data heterogeneity are present. Under this setting, clients with small compute budgets often experience large performance degradation. Our goal is to enable clients with limited model sizes to obtain competitive performance. Although model generation, submodel extraction, and prototype-based alignment have each been studied individually, none of them alone directly addresses this objective in such a setting.
>
> In FRAMP, **these techniques do not operate independently**. They work in a coordinated manner, and the integration introduces several non-trivial challenges:
>
> 1. The hypernetwork generates personalized full models, and these models should be compatible with extraction into multiple size-constrained submodels. This requires a stable parameter structure and a consistent basis for importance ranking across clients.
>
> 2. Submodel extraction can disrupt representation consistency. Prototype-based alignment is therefore needed to maintain comparable representations across heterogeneous model capacities.
>
> 3. Prototype alignment relies on the hypernetwork to generate personalized models with parameterizations are still comparable across clients, ensuring prototypes remain meaningful.
>
> These modules complement and reinforce each other within FRAMP. The hypernetwork provides personalized full-models, which enable submodel extraction to offer compute-constrained submodels tailored to each client. Prototype alignment ensures these extracted models learn representations that remain coherent and comparable across different model sizes. This alignment works because the hypernetwork produces personalized models whose parameterizations remain compatible across clients.
>
> Through this interdependence, FRAMP forms a unified mechanism that allows clients with diverse data and computation budgets to obtain significantly more balanced performance.
>
> > W2: Do high structural similarity and reduced diversity among submodels bring any crucial limitations? Any evidence? To support the claim that more activations on the early portion of model parameters, the submission should fix the total model size and only change the activation locations (e.g., early portion → mid portion → late portion) rather than change the model size and activation locations simultaneously.
>
> Thank you for the comment. To ensure clarity, we first outline what Fig. 1 implies. Fig. 1 visualizes how frequently each parameter is selected by clients within the **same model-size group**. When many clients select overlapping subsets of parameters, the cumulative coverage curve rises quickly in regions of frequently selected parameters and remains nearly flat in regions where parameters are rarely used. This behavior reflects high cross-client structural similarity and under-utilization of a large portion of the parameters in FIARSE, rather than differences between early, mid, or late layers.
>
> Fig. 2 shows the consequence of this pattern. Since clients tend to rely on the same subset of parameters, and many parameters receive limited training, the resulting global backbone fails to provide submodels that work uniformly well for all clients. As shown in Fig. 2, there is substantial performance variance within model-size group and large performance gaps between groups, especially for clients with small compute budgets. These observations directly show that **high structural similarity and reduced parameter diversity lead to uneven performance** under system and data heterogeneity.
>
> FRAMP generates personalized full models and extracts submodels that better exploit the parameter space. This results in considerably more balanced performance across clients of different model sizes. The corresponding plots for FRAMP are provided in Fig. 5(b) and in Fig. 8 of Appendix D.7. This improvement provides direct evidence that reducing cross-client structural similarity and avoiding concentration on a single parameter subset leads to more uniform and reliable performance under heterogeneous compute constraints.
>
> > W3: Better to include more results of the overhead on additional data transmissions.
>
> Thank you for the comment. Compared to standard heterogeneous FL baselines, FRAMP introduces only two small additional data transmissions. (i) At initialization, each client uploads its descriptor once. A 128-dimensional descriptor is roughly 0.5 KB. (ii) During training, the only extra cost is the exchange of class prototypes each communication round. For ResNet18 on CIFAR10, a full set of prototypes (10 classes × 512 dimensions) is about 20 KB. We have added these overheads in Appendix D.5 of the revised manuscript.

---

> ### Author Response · Authors · 2025-11-22
> **2/2 - Rebuttal by Authors**
>
> > W4: Though the Appendix provides the experimental results with Dirichlet-\alpha = 0.1, it is better to include more non-IID configurations.
>
> Thank you for the suggestion. We have added experiments under both stronger and milder non-IID conditions ($\alpha=0.05$ and $\alpha=0.5, 0.7$). The results in Table 1-3 show that FRAMP consistently maintains more balanced client performance across these additional heterogeneity levels. The new results have been added into Appendix D.2 of the updated manuscript.
>
> We also note that our evaluation already includes other forms of heterogeneity, including system heterogeneity (five submodel-size groups) and 20\% client participation. Together, these experiments provide a broader assessment of FRAMP under diverse heterogeneous FL settings.
>
> **Table 1: Test accuracy on CIFAR100 with stronger data heterogeneity ($\alpha = 0.05$).**
>
> | **Method** | **Local** | **1/64** | **1/16** | **1/4** | **1.0** | **Union** |
> |-------------|:---------:|:--------:|:--------:|:--------:|:--------:|:---------:|
> | **HeteroFL** | 22.24 | 17.28 | 21.96 | 22.16 | 27.56 | 21.48 |
> | **FedRolex** | 12.63 | 3.08 | 5.84 | 16.12 | 25.48 | 11.67 |
> | **ScaleFL**  | 29.14 | 28.68 | 32.04 | 28.64 | 27.20 | 27.71 |
> | **FIARSE**   | 29.29 | 25.68 | 31.84 | **29.24** | 30.40 | 29.25 |
> | **FRAMP**    | **30.92** | **31.32** | **33.12** | 28.72 | **30.52** | **30.10** |
>
> **Table 2: Test accuracy on CIFAR100 with milder data heterogeneity ($\alpha = 0.5$).**
>
> | **Method**   | **Local** |  **1/64** |  **1/16** |  **1/4**  |  **1.0**  | **Union** |
> | ------------ | :-------: | :-------: | :-------: | :-------: | :-------: | :-------: |
> | **HeteroFL** |   30.73   |   25.64   |   28.36   |   33.16   |   35.76   |   29.18   |
> | **FedRolex** |   19.06   |    3.28   |   10.52   |   27.28   |   35.16   |   18.81   |
> | **ScaleFL**  |   40.64   |   35.48   |   43.16   |   43.20   |   40.72   |   37.30   |
> | **FIARSE**   |   41.13   |   37.88   |   43.08   |   43.80   |   39.76   |   38.91   |
> | **FRAMP**    | **42.22** | **40.96** | **43.20** | **43.88** | **40.84** | **40.17** |
>
>
> **Table 3: Test accuracy on CIFAR100 with milder data heterogeneity ($\alpha = 0.7$).**
>
> | **Method**   | **Local** |  **1/64** |  **1/16** |  **1/4**  |  **1.0**  | **Union** |
> | ------------ | :-------: | :-------: | :-------: | :-------: | :-------: | :-------: |
> | **HeteroFL** |   31.16   |   28.60   |   28.80   |   31.40   |   35.84   |   29.69   |
> | **FedRolex** |   19.06   |    3.12   |   11.76   |   26.80   |   34.56   |   18.80   |
> | **ScaleFL**  |   42.05   |   39.96   |   43.28   | **43.44** | **41.52** |   38.44   |
> | **FIARSE**   |   40.31   |   38.20   |   41.44   |   41.24   |   40.36   |   37.71   |
> | **FRAMP**    | **43.25** | **44.84** | **44.32** |   42.72   |   41.12   | **40.53** |

---

> ### Comment · Reviewer_MSoF · 2025-11-25
>
> Thanks for authors’ efforts on preparing rebuttals. My concerns are addressed. I will keep my current positive score.

---

> > ### Author Response · Authors · 2025-11-25
> >
> > We are glad that our responses addressed your concerns. Thank you again for your helpful comments.

---

### Official Review · Reviewer_fzVX · 2025-11-09

**Soundness:** 2
**Presentation:** 3
**Contribution:** 2
**Rating:** 2
**Confidence:** 3

**Summary:**

The paper asks two questions of the personalized Federated Learning (pFL) setup - a) for each client in a communication round, how to construct personalized submodel that adapts to both computational constraints and data distribution, and b) how to promote semantic consistency across clients in the absence of any shared reference dataset? The paper develops a new framework called FRAMP which is designed to incorporate an affirmative answer to both of these questions. FRAMP uses a hypernetwork on the server to instantiate personalized full-size model for each client, and thresholds model weights based on magnitude to achieve a target number of non-zero weights. Experimental evidence is provided for improved statistical performance compared to baselines.

**Strengths:**

(S1) The writing and presentation in the paper is clear. The problem is well motivated and FRAMP's design is explained well. Prior work is well cited although contextualization could be improved further.

**Weaknesses:**

(W1) Lines 284-287 state that each participating client updates all parameters $\omega_n$. This defeats the purpose of constructing personalized submodels that adapt to computational constraints (Line 50, Q1). Further, there isn't any theoretical/empirical study in the paper that shows FRAMP's advantage in terms of computational constraints.

(W2) It is difficult to justify originality and significance when contextualized against [FedDSE] (missing citation; relevant prior work) and [PeFLL]. FedDSE seems to address the two problems a) Limited Structural Diversity, and b) Client-Agnostic Importance Estimation, that are mentioned in section 3.2 as drawbacks of FIARSE.

[FedDSE] Haozhao Wang, Yabo Jia, Meng Zhang, Qinghao Hu, Hao Ren, Peng Sun, Yonggang Wen, and Tianwei Zhang. 2024. "FedDSE: Distribution-aware Sub-model Extraction for Federated Learning over Resource-constrained Devices". In Proceedings of the ACM Web Conference 2024 (WWW '24), 2902–2913. ACM. DOI: 10.1145/3589334.3645416.

(W3) Loose statements have been made at several places in the paper -
- Lines 252-254: "... jointly learn the mask and form the submodel, ..." This is somewhat misleading. As best I can tell, equation (3) is implemented as optimize weights and then threshold, which is different from jointly optimizing.
- Lines 363-365: "Fig. 5c demonstrates representation alignment, with local prototypes initially dispersed (left, early stage) converging toward global prototypes (right, later stage), leading to tighter clusters and improved consistency across clients." It is not clear to me how Figure 5c is conveying this statement.
- Line 140: "Some recent methods incorporate importance scores to guide mask selection." Which methods? Citation is missing.
- Line 176-177: "... clients learn inconsistent class prototypes due to model heterogeneity and non-IID data, ultimately compromising generalization performance." Where is this established? Citation? By itself, differences in steady state client accuracies under FIARSE (Figure 2) is not evidence that personalization is not working well. Optimal accuracies can indeed vary across clients.
- Line 439-440: "We attribute this to the difficulty of learning structured sparsity and controlling sparsity levels." This is a hypothesis, not an attribution.

(W4) Line 485: While generalization testing has been explained in sections 5.3 and 5.4, personalization testing is inadequately explained/referenced.

Things to improve the paper that did not impact the score:
- Figure 4: This is better represented as a cdf type plot. $F(a)$: num clients with test accuracy $>= a$. One plot for each size value, and each plot has all algorithms of interest.

**Questions:**

(Q1) Lines 155-159: While this is an interesting observation, there could be an alternate explanation for this, viz. the early layers are more about generalization and learning shared features than about personalization.

(Q2) Could the authors address (W1) and (W2)?

---

> ### Author Response · Authors · 2025-11-22
> **1/3 - Rebuttal by Authors**
>
> We thank the reviewer for the detailed feedback, which provided opportunities to clarify several important aspects of the method.
>
> > W1: Lines 284-287 state that each participating client updates all parameters. FRAMP's advantage in terms of computational constraints.
>
> Thank you for pointing out this ambiguity, which originated from a notation typo in Line 284–287. We have revised the notation to explicitly clarify that **each client updates only the submodel parameters**.
>
> For the second question, in our framework, 'computational constraints' denotes predefined per-client model capacity budgets. **Each client strictly follows its capacity** and trains a submodel of that size. Our goal is to ensure that clients with tight budgets (i.e., small submodels) do not suffer severe performance degradation compared to those with larger budgets.
>
> Our empirical study directly **compares all methods under identical model-size budgets settings**, and results consistently demonstrate FRAMP’s advantages:
>
> 1. Section 5.2: Clients with small submodels (e.g., 1/64 model size) achieve substantially higher accuracy under FRAMP than other competing baselines.
>
> 2. Section 5.3: When clients request submodels of other sizes, FRAMP can generate such submodels that still obtain comparable performance without retraining.
>
> 3. Section 5.4: New clients with unseen data also benefit from FRAMP’s personalized model generation and maintain strong performance.
>
> Together, these results demonstrate that FRAMP yields **more balanced and better performance across different model-size groups**, especially for clients with the strictest compute budgets, and outperforms baselines under the same computational constraints.
>
>
> > W2:  It is difficult to justify originality and significance when contextualized against [FedDSE] (missing citation; relevant prior work) and [PeFLL].
>
> Thank you for raising the connection to FedDSE and PeFLL. We have cited FedDSE on Line 39 and 766. PeFLL was already included in the original submission.
>
> Although FedDSE considers neuron-activation differences across clients, the problem setting and solution mechanism are fundamentally different from FRAMP:
>
> 1. **No shared full-size backbone in FRAMP.** FedDSE still relies on a single global full-size model, from which clients extract neurons. In contrast, FRAMP replaces the global backbone entirely with a hypernetwork that generates a personalized full model for each client. Thus, the "competition for shared neurons" highlighted by FedDSE **does not occur** in FRAMP, and structural diversity is substantially higher than in FedDSE.
>
> 2. **Server-side submodel extraction in FRAMP.** FedDSE requires sending the full model to clients for local neuron selection based on inference activations.
> However, this design assumes that all clients are capable of loading and processing the full model, which may not hold under heterogeneous resource constraints where some clients have very limited capacity.
> FRAMP performs submodel extraction on the server side, so clients receive only their lightweight submodels.
>
> 3. **Different extraction mechanisms and objectives.** FedDSE performs layer-wise selection based on neuron activations during inference. FRAMP instead uses parameter magnitude-based masks. Our ablation study in Section 5.5 compares alternative extraction strategies and demonstrates the effectiveness of our approach.
>
> **Thus, although FedDSE partially relates to the issues we discuss for FIARSE, it is not designed to address these limitations, and its assumptions and mechanisms are orthogonal to FRAMP.**
>
> Regarding PeFLL, it is a personalized federated learning method focusing on data heterogeneity. It does not support heterogeneous computational budgets and therefore cannot be directly applied in the model-heterogeneous setting.

---

> ### Author Response · Authors · 2025-11-22
> **2/3 - Rebuttal by Authors**
>
> > W3 - 1: Lines 252-254: "... jointly learn the mask and form the submodel, ..." This is somewhat misleading.
>
> Thank you for pointing out this possible ambiguity. Our intention was not to imply that the mask is explicitly parameterized or optimized. Rather, the mask is implicitly induced by the parameter magnitudes, as formalized in Eq. (3). Since the mask is a deterministic function of the model parameters, any update to $\omega_n$ automatically changes the resulting mask. In this sense, the submodel is **implicitly coupled** with weight optimization, without introducing additional trainable mask parameters.
>
> We have revised the text to "optimizing the model parameters implicitly determines the mask" to avoid potential confusion.
>
> > W3 - 2: It is not clear to me how Figure 5c is conveying this statement.
>
> Each dot in Fig. 5(c) represents a local prototype computed independently by each client (one per class per client), while the stars denote the global prototypes aggregated at the server. Thus, for each class (color), the figure visualizes 100 local prototypes from 100 clients.
>
> In the early stage (left), local prototypes of the same class are relatively dispersed and tend to form several small clusters (around 4 in this figure). In the later stage (right), these local prototypes cluster more tightly around the corresponding global prototype. This illustrates the effect of prototype alignment, where clients’ class-level representations become more consistent and better aligned with the global prototypes over training.
>
> We have refined the corresponding description in the manuscript to present this visualization more clearly.
>
> > W3 - 3: Line 140 Citation is missing.
>
> We have updated the citation.
>
> > W3 - 4: Line 176-177. Where is this established? Citation? By itself, differences in steady state client accuracies under FIARSE (Figure 2) is not evidence that personalization is not working well. Optimal accuracies can indeed vary across clients.
>
> We clarify that the statement “clients learn inconsistent class prototypes due to model heterogeneity and non-IID data” is not derived from Fig. 2, but is supported by prior prototype-based FL literature. We have added the corresponding citation in the manuscript.
>
> Fig. 2 illustrates that clients with smaller compute budgets show consistently and substantially lower accuracy than other clients, and the variation within the same model-size group is larger. While some performance variation across clients is natural under non-IID data, this systematic underperformance within small model-size group suggests that the extracted submodels do not sufficiently adapt to client-specific data needs under compute constraints. This motivates the design of FRAMP, which aims to provide more balanced performance across heterogeneous clients.
>
> > W3 - 5: Line 439-440. This is a hypothesis, not an attribution.
>
> We clarify that the statement in Lines 439–440 refers to the MLP-based submodel extraction strategy evaluated in our ablation study, rather than to our proposed method.  The intention was to explain the observed instability of this strategy, rather than a causal attribution. We have refined the phrasing in the manuscript to reflect this.
>
> > W4: Line 485. While generalization testing has been explained in sections 5.3 and 5.4, personalization testing is inadequately explained/referenced.
>
> The line referenced by the reviewer appears in the conclusion, and we have clarified it in the updated manuscript.
>
> Our personalization evaluation corresponds to the main results in Section 5.2, which report per-client performance under heterogeneous data and compute budgets. Table 1 provides the accuracy averaged within each model-size group as well as over all clients. Fig. 4 and Fig. 8 visualize the distribution of client accuracies across groups. Fig. 5(b) illustrates a more diverse pattern of parameter activations under FRAMP, supporting better adaptation to heterogeneous clients. Together, these results show that FRAMP delivers substantially more balanced performance across clients, especially for those with smaller submodel sizes, which reflects stronger personalization.

---

> ### Author Response · Authors · 2025-11-22
> **3/3 - Rebuttal by Authors**
>
> > Q1: Lines 155-159: While this is an interesting observation, there could be an alternate explanation for this, viz. the early layers are more about generalization and learning shared features than about personalization.
>
> Thank you for the insightful question. To avoid misunderstanding, we first clarify what Fig. 1 implies.
>
> For each model-size group (1.0, 1/4, 1/16, 1/64), we collect the masks produced by clients in that group and compute the average activation probability for each parameter. We then plot the cumulative coverage, which shows the cumulative proportion of activated parameters covered up to a given parameter index. **The curve reflects how often each parameter is selected by clients within the same model-size group.**
>
> If masks from different clients are diverse, each parameter should have a similar chance of being selected. In this case, the curve is close to a straight diagonal line, similar to that for the size = 1.0 case. If many clients select overlapping subsets of parameters, the curve rises quickly in regions where parameters are frequently selected and becomes nearly flat in regions where parameters are rarely used. This pattern clearly appears in the size = 1/16 and 1/64 groups.
>
> FIARSE exhibits this concentrated behavior. Clients within the same model-size group tend to activate highly overlapping subsets of parameters, while a large portion of parameters show very low activation probability. This arises from the shared global importance ranking, rather than from any assumption about the semantic roles of early or late layers.
>
> Even if early layers tend to contain more general features, this does not explain why clients with **heterogeneous** data distributions select almost the same parameter ranges, nor why many parameters receive almost no training. The key observation is the high cross-client mask overlap, which indicates reduced structural diversity and under-utilization of model capacity.
>
> We have slightly refined the corresponding paragraph in the manuscript to make the intended meaning more explicit.

---

> ### Author Response · Authors · 2025-11-28
>
> We thank the reviewer again for the earlier detailed review. As the discussion phase approaches its end, we would like to confirm whether our clarifications have addressed the raised concerns. We are happy to provide further elaboration should any additional questions remain.

---

### Author Response · Authors · 2025-12-01
**Summary by Authors**

Dear AC,

Thank you for your efforts in overseeing the review and discussion process. For convenience, we summarize the review outcomes and key contributions below.

---

**Review Summary**

**3** of the **4** reviewers now express a **positive** stance toward the submission and recognize the technical value of the proposed FRAMP framework. Reviewer **MSoF**, **SYMr**, and **FCqt** indicated that our clarifications have addressed their concerns. Reviewer **SYMr** wrote on **Nov 24** that **"I raise the score to 6"**. These resolved concerns include the interpretation of structural similarity, communication and computation overhead, additional non-IID and IID experiments, privacy discussion, and the rationale behind the parameter extraction strategy.


The remaining reviewer, **fzVX**, raised several points requiring clarification in the review but **did not** re-engage during the discussion window up to **Nov 27**. All points raised were carefully addressed in the rebuttal, including the update process, relation to FedDSE, interpretation of Fig. 1, and several explanation details. These points mainly stem from presentation ambiguities rather than methodological issues, and do not affect the validity or conclusions of the proposed method.

---

**Main Contributions**

* **A unified framework for personalized and resource-adaptive federated learning**, jointly addressing system and data heterogeneity, enabling clients with different computational budgets and data distributions to obtain effective personalized submodels.

* **Extensive experiments under diverse heterogeneity settings** (varying model-size budgets, non-IID levels, unseen clients, unseen budgets, and multiple ablations) consistently show improved personalization and generalization, especially for clients with limited computation budgets.

---

We sincerely thank the AC for the time and effort on this work, and hope that our clarifications and the reviewers’ updated opinions will help inform the AC’s final assessment.


Best regards,

Authors

---

### Meta-Review · Area_Chair_DgFe · 2026-01-07

**Summary:**

The reviewers'concerns are mostly focusing on 1) its novelty and originality. There are existing prior works that are similar, and the components of the proposed method, including Hypernetworks, sub-model extraction, and prototype alignment, are not novel.  2) Methodological Rigor. There are ambiguity in multiple definitions ("client descriptor", "local prototypes" etc), concerns over the assumption that only "high-value" parameters are optimal, what the "importance threshold" actually represents. 3. Insufficient analysis. The privacy discussion is only qualitative and fails to address risks like gradient-inversion attacks. The analysis on efficiency and overhead is also not sufficient.

**Reviewer Concerns:**

Most of concerns over the missing citations, insufficient analysis and the methodological rigor are addressed by the rebuttal. Yet, I believe issue with the novelty and originality is still outstanding based on the following concerns : 1) each component of the proposed method, including Hypernetworks, sub-model extraction, and prototype alignment, is not novel, as pointed by reviewers, which makes the contribution of proposed method incremental. 2) Insufficient comparison on recent works/baselines. Reviewer fzVX pointed out missing related works such as FedDSE. In addition, there have been a growing number of FL works to address model heterogeneity and data heterogeneity, such as [1-3] and some recent works discussed in [2-3], e.g. FedTGP. The paper only has 3 references from 2024-2025, which does not provide a comprehensive context for the proposed method. It is suggested to include a more comprehensive discussion on the related works, and perform experimental comparisons to justify the superiority of proposed framework, against other recent methods that address the same goal of system and data heterogeneity.

1. Deng et al, TailorFL: Dual-Personalized Federated Learning under System and Data Heterogeneity, 2022
2. Chen et al, Advances in Robust Federated Learning: A Survey With Heterogeneity Considerations 2025
3. Zhang et al, HtFLlib: A Comprehensive Heterogeneous Federated Learning Library and Benchmark 2025

**Reviewer Scores:**

During rebuttal, only one reviewer (fzVX) didn't participate fully in the discussion.  I think the reviewer is likely to maintain or raise his score but it may still be negative given that the novelty issue is still outstanding.

---

### Decision · Program_Chairs · 2026-01-26

Reject